**A pulse-decay method for low permeability analyses of granular rock**
**media**
Tao Zhang[1], Qinhong Hu[1,2*], Behzad Ghanbarian[3], Derek Elsworth[4], Zhiming Lu[5]
[1] Department of Earth and Environment Sciences, University of Texas at Arlington,
Arlington, TX 76019, United States
[2]National Key Laboratory of Deep Oil and Gas, China University of Petroleum (East
China), Qingdao 266580, P.R. China
[3] Porous Media Research Lab, Department of Geology, Kansas State University, Manhattan,
KS 66506, United States
[4] Department of Energy and Mineral Engineering, G3 Centre and Energy Institute, The
Pennsylvania State University, University Park, PA 16802, United States
[5] The Earth and Environmental Sciences Division, Los Alamos National Laboratory, Los
Alamos, NM 87544, United States
*Accepted for*
*Hydrology and Earth System Sciences*
*Oct 20, 2023*
* Corresponding author: huqinhong@upc.edu.cn

**Abstract:** Nano-darcy level permeability measurements of porous media, such as nano-porous mudrocks, are frequently conducted with gas invasion methods into granular-sized samples with short diffusion lengths and thereby reduced experimental duration; however, these methods lack rigorous solutions and standardized experimental procedures. For the first time, we resolve this by providing an integrated technique (termed as gas permeability technique) with coupled theoretical development, experimental procedures, and data interpretation workflow. Three exact mathematical solutions for transient and slightly compressible spherical flow, along with their asymptotic solutions, are developed for early- and late-time responses. Critically, one late-time solution is for an ultra-small gas-invadable volume, important for a wide range of practical usages. Developed as applicable to different sample characteristics (permeability, porosity, and mass) in relation to the storage capacity of experimental systems, these three solutions are evaluated from essential considerations of error difference between exact and approximate solutions, optimal experimental conditions, and experimental demonstration of mudrocks and molecular-sieve samples. Moreover, a practical workflow of solution selection and data reduction to determine permeability is presented by considering samples with different permeability and porosity under various granular sizes. Overall, this work establishes a rigorous, theory-based, rapid,

and versatile gas permeability measurement technique for tight media at sub-

nano darcy levels.

**Keywords:** permeability; granular samples; pulse-decay; mathematical

solutions; experimental methods.

**Highlights:**

- An integrated (both theory and experiments) gas permeability technique (GPT) is presented.
- Exact and approximate solutions for three cases are developed with error discussion.
- Conditions of each mathematical solution are highlighted for critical parameters.
- Essential experimental methodologies and data processing procedures are provided and evaluated.

## 1. Introduction

Shales, crystalline, and salt rocks with low permeabilities (e.g., $<10^{-17}$ m$^2$ or 10 micro-darcies $\mu$D) are critical components to numerous subsurface studies. Notable examples are the remediation of contaminated sites(Neuzil, 1986; Yang et al., 2015), long-term performance of high-level nuclear waste repositories (Kim et al., 2011; Neuzil, 2013), enhanced geothermal systems (Huenges, 2016; Zhang et al., 2021), efficient development of unconventional oil and gas resources (Hu et al., 2015; Javadpour, 2009), long-term sealing for carbon utilization and storage (Fakher et al., 2020; Khosrokhavar, 2016), and high-volume and effective gas (hydrogen) storage (Liu et al., 2015; Tarkowski, 2019). For fractured rocks, the accurate characterization of rock matrix and its permeability is also critical for evaluating the effectiveness of low-permeability media, particularly when transport is dominated by slow processes like diffusion (Ghanbarian et al., 2016; Hu et al., 2012).

Standard permeability test procedures in both steady-state and pulse-decay methods use consolidated cm-sized core-plug samples, which may contain fractures and show dual- or triple-porosity characteristics (Abdassah and Ershaghi, 1986; Bibby, 1981). The overall permeability may therefore be controlled by a few bedding-oriented or cross-cutting fractures, even if experiments are conducted at reservoir pressures (Bock et al., 2010;

Gensterblum et al., 2015; Gutierrez et al., 2000; Luffel et al., 1993). Fractures might be naturally- or artificially-induced (e.g., created during sample processing), which makes a comparison of permeability results among different samples difficult (Bock et al., 2010; Gensterblum et al., 2015; Gutierrez et al., 2000; Luffel et al., 1993). Hence, methods for measuring the matrix (non-fractured) permeability in tight media, with a practical necessity of using granular samples, have attracted much attention to eliminate the sides effect of fractures (Civan et al., 2013; Egermann et al., 2005; Heller et al., 2014; Wu et al., 2020; Zhang et al., 2020).

A GRI (Gas Research Institute) method was developed by Luffel et al. (1993) and followed by Guidry et al. (1996) to measure the matrix permeability of crushed mudrocks (Guidry et al., 1996; Luffel et al., 1993). Such a method makes permeability measurement feasible in tight and ultra-tight rocks (with permeability $< 10^{-20}$ m$^2$ or 10 nano-dcarcies, nD), particularly when permeability is close to the detection limit of the pulse-decay approach on core plugs at ~10 nD (e.g., using commercial instrument of PoroPDP-200 of CoreLab). In the GRI method, helium may be used as the testing fluid to determine permeability on crushed samples at different sample sizes (e.g., within the 10-60 mesh range, which is from 0.67 mm to 2.03 mm). The limited mesh size of 20-35 (500-841 μm in diameter) was recommended in earlier

works, which has led to the colloquial names of "the GRI method/size" in the literature (Cui et al., 2009; Kim et al., 2015; Peng and Loucks, 2016; Profice et al., 2012). However, Luffel et al. (Guidry et al., 1996; Luffel et al., 1993) did not document the processing methodologies needed to derive the permeability from experimental data from such a GRI method. That is, there are neither standard experimental procedures for interpreting gas pulse-decay data in crushed rock samples nor detailed mathematical solutions available for data processing in the literature (Kim et al., 2015; Peng and Loucks, 2016; Profice et al., 2012). In this work, we achieve to: (1) develop mathematical solutions to interpret gas pulse-decay data in crushed rock samples without published algorithm available as this method shares different constitutive phenomena to the traditional pulse-decay method for core plug samples in Cartesian coordinates; and (2) present associated experimental methodology to measure permeability, reliably and reproducibly, in tight and ultra-tight granular media.

We first derive the constitutive equations for gas transport in granular (unconsolidated or crushed rock) samples. Specifically, we develop three mathematical solutions which cover different experimental situations and sample properties. As each solution shows its own pros and cons, we then in detail present the error analyses for the derived exact and approximate

solutions and discuss their applicable requirements and parameter
recommendation for practical usages. This work aims to fill the knowledge
gap of the granular rock (matrix) permeability measurement and follow-on
literature by establishing an integrated methodology for reproducible
measurements of nD-level permeability in tight rock for emerging energy and
resources subsurface studies.

## 2. Mathematical solutions for gas permeability of granular samples

For a compressible fluid under unsteady-state conditions, flow in a porous
medium can be expressed by the mass conservation equation:
$$\frac{\partial p}{\partial t} + \nabla \cdot (\rho \overline{v}) = 0 \qquad (1A)$$

where $p$ is the pressure, $t$ is the time, $\rho$ is the fluid density, and $\overline{v}$ is the
Darcy velocity. In continuity equations derived for gas flow in porous media,
permeability can be treated as a function of pressure through the ideal gas law.
Constitutive equations are commonly established for a small pressure
variation to avoid the non-linearity of gas (the liquid density to be a constant)
and to ensure that pressure would be the only unknown parameter (Haskett et
al., 1988). For spherical coordinates of fluid flow in porous media, assuming
flow along the radial direction of each spherical solid grain, Eq. (1A) becomes
$$\frac{\partial p}{\partial t} \phi = \frac{1}{c_t} \frac{k}{\mu r^2} \frac{\partial}{\partial r} \left( r^2 \frac{\partial p}{\partial r} \right) \qquad (1B)$$

The gas compressibility $c_t$ is given by

$$c_t = \frac{1}{\rho}\frac{d\rho}{dp} = \frac{1}{p} - \frac{1}{z}\frac{dz}{dp}$$                            (1C)

In Eqs. (1B) and (1C), $\phi$ and $k$ are sample porosity and permeability, $r$
is the migration distance of fluid, $\mu$ is the fluid viscosity, and $z$ is the gas
deviation (compressibility) factor and is constant.
To correct for the non-ideality of the probing gas, we treat gas density as a
function of pressure and establish a relationship between the density and the
permeability through a pseudo-pressure variable (given in the 1[st] part of
Supplemental Information SI1). Detailed steps for deriving mathematical
solutions for the GPT can be found in SI2, based on heat transfer studies
(Carslaw and Jaeger, 1959). The Laplace transform is an efficient tool for
solving gas transport in granular samples with low permeabilities, as applied
in this study. Alternatively, other approaches, such as the Fourier analysis,
Sturm-Liouville method, or Volterra integral equation of the second form may
be used (Carslaw and Jaeger, 1959; Haggerty and Gorelick, 1995; Ruthven,

1984).

We applied dimensional variables to derive the constitutive equation given
in Eq. (S10) for which the initial and boundary conditions are

$$\frac{\partial^2 U_s}{\partial \xi^2} + s^2 U_s = 0 \Big|_{U_s=0, \xi=0} \qquad (2A)$$

$$\alpha^2 (U_s - 1) = \frac{3}{K_c} \left( \frac{\partial U_s}{\partial \xi} - \frac{U_s}{\xi} \right) \Big|_{\xi=1} \qquad (2B)$$

where $U_s$ and $\xi$ represent the dimensionless values of gas density and sample scale, and $s$ is the transformed Heaviside operator. $\alpha$ in Eq. (2B) is determined by solving Eq. (S30) for its root. $K_c$ in Eq. (2B) is a critical parameter that represents the volumetric ratio of the total void volume of the sample cell to the pore volume of the porous samples. It is similar to the storage capacity, controlling the acceptable measurement range of permeability and decay time, in the pulse-decay method proposed by Brace et al. (1968).

The fractional gas transfer for the internal (limited $K_c$ value) and external (infinite $K_c$ value) gas transfer of sample is given by

$$F_f = 1 - 6 \sum_{n=1}^{\infty} \frac{K_c(1+K_c)e^{-\alpha_n^2 \tau}}{9(K_c+1) + \alpha_n^2 K_c^2} \qquad (2C)$$

$$F_s = 1 - \frac{6}{\pi^2} \sum_{n=1}^{\infty} \frac{e^{-(n\pi)^2 \tau}}{n^2} \qquad (2D)$$

where $F_f$ and $F_s$ represent the uptake rate of gas outside and inside the sample separately as a dimensionless parameter, and $\tau$ is the Fourier number of dimensionless time. Three approximate solutions of the transport

coefficient based on Eqs. (2C) and (2D) for various conditions are presented
below.
The late-time solution to Eq. (2C) for a limited $K_c$ value (called LLT
hereafter) is
$$k = \frac{R_a{}^2 \mu c_t \phi_f s_1}{\alpha_1{}^2} \qquad (3A)$$

The late-time solution to Eq. (2D) when $K_c$ tends to infinity (ILT hereafter)
is
$$k = \frac{R_a{}^2 \mu c_t \phi_f s_2}{\pi^2} \qquad (3B)$$

The early-time solution to Eq. (2D) when $K_c$ approaches infinity (IET
hereafter) is
$$k = \frac{\pi R_a{}^2 \mu c_t \phi_f s_3}{36} \qquad (3C)$$

In Eq. (3), $R_a$ is the particle diameter of a sample, and $s_1$, $s_2$, and $s_3$ are
the three exponents that may be determined from the slopes of data on double
logarithmic plots. Table 1 summarizes Eqs. (3A) to (3C) and conditions under
which such approximate solutions would be valid.
Table 1. Solutions schematic with difference $K_c$ and $\tau$ values

| Parameter | Symbol | Remarks | | |
|---|---|---|---|---|
| Volume fraction§ | $K_c$ | Limited value for $K_c < 10$ | Infinity value for $K_c > 10$ | |
| Exact. Density fraction£ | $F$ | $F_f$ | $F_s$ | |
| Approx. Solution of Density fraction* | Eqs. (3A-3B) | Eq. (3A) (LLT) | Eq. (3C) (IET) | Eq. (3B) ) (ILT) |
| Available Dimensionless time for Approx. solution | $\tau$ | Late-time solution $\tau > 0.024$ | Early-time solution $\tau < 0.024$ | Late-time solution $\tau > 0.024$ |

§ It defines as the volumetric ratio of the total void volume of the sample cell to the pore volume of the porous samples, the classification between the limited and infinity value is proposed as 50 with the following analyses.

£ The original constitutive equation for different $K_c$ value.
* Eqs. (3A-3C) are three approximate solutions of density faction function $F$.

185  Based on diffusion phenomenology, Cui et al. (2009) presented two

186 mathematical solutions similar to our Eqs. (3A) and (3C). In the work of Cui

187 et al. (2009), however, one of late-time solution is missing, and error analyses

188 are not provided. Besides, the lack of detailed analyses of $\tau$ and $K_c$ in the

189 constitutive equations will likely deter the practical application of Eq. (3B),

190 which is able to cover an experimental condition of small sample mass with a

191 greater $\tau$ (further analyzed in Section 3). Furthermore, the early-time and

192 late-time solution criteria are not analyzed, and the pioneering work of Cui et

193 al. (2009) does not comprehensively assess practical applications of their two

194 solutions in real cases, which is addressed in this study. Hereafter, we refer to

195 the developed mathematical and experimental, gas-permeability-measurement

196 approach holistically as gas permeability technique (GPT).

197 **3. Practical usages of algorithms for the GPT**

198  As aforementioned, mathematical solutions given in Eqs. (3A) and (3B)

199 were deduced based on different values of $K_c$ and $\tau$ as shown in the SI2.

200 This means each solution holds only under specific experimental conditions,

201 which are mostly determined by the permeability, porosity, and mass of

202 samples, as well as gas pressure and void volume of the sample cell. In this

203 section, the influence of parameters $K_c$ and $\tau$ on the solution of constitutive

equation is analyzed and a specific value of dimensionless time ($\tau = 0.024$) is
proposed as the criterion required to detect the early-time regime from the late-
time one for the first time in the literature. We also demonstrate that the early-
time solution of Eq. (3C), which has been less considered for practical
applications in previous studies, is also suitable and unique under common
situations. Besides, the error of the approximate solution compared to the
exact solution and their capabilities are discussed, as it helps to select an
appropriate mathematical solution at small $\tau$ values. Moreover, we showcase
the unique applicability and feasibility of the new solution of Eq. (3B).
**3.1 Sensitivity analyses of the $K_c$ value for data quality control**
To apply the GPT method, appropriately selecting the parameter $K_c$ in Eqs.
(3A)-(3C) is crucial, as it is a critical value for data quality control. The
dimensionless density outside the sample, $U_f$, is related to $K_c$ via Eq. (S33)
in the SI2. One may simplify Eq. (S33) by replacing the series term with some
finite positive value and set
$$U_f - \frac{K_c}{1+K_c} > 0 \qquad\qquad (1G)$$

We define $K_f = K_c/(1 + K_c)$ to interpret the density variance of the system
as $K_f$ is closely related to the dimensionless density outside the sample, $U_f$.
Eq. (1G) shows the relationship between the $U_f$ and $K_c$ (Fig. 1). For
$K_c > 0$, $K_f$ falls between 0 and 1. The greater the $K_f$ value is, the insensitive
to density changes the system would be. For $K_c$ equal to 50, $K_f$ would no
longer be sensitive to $K_c$ variations as it has already approached 98% of the
dimensionless density. This means that the $U_f$ value needs to be greater than
0.98, and this leaves only 2% of the fractional value of $U_f$ available for
capturing gas density change. When $K_c$ is 100, the left fractional value of $U_f$
would be 1%. This would limit the amount of data available (the linear range
in Fig. S1) for the permeability calculation, which would complicate the data
processing. Thus, for the GPT experiments, a small value of $K_c$ (less than 10)
is recommended, as $K_f$ nearly reaches its plateau beyond $K_c = 10$ (Fig. 1).
When $K_c$ is 10, the left fractional value of $U_f$ is only as low as 9%.

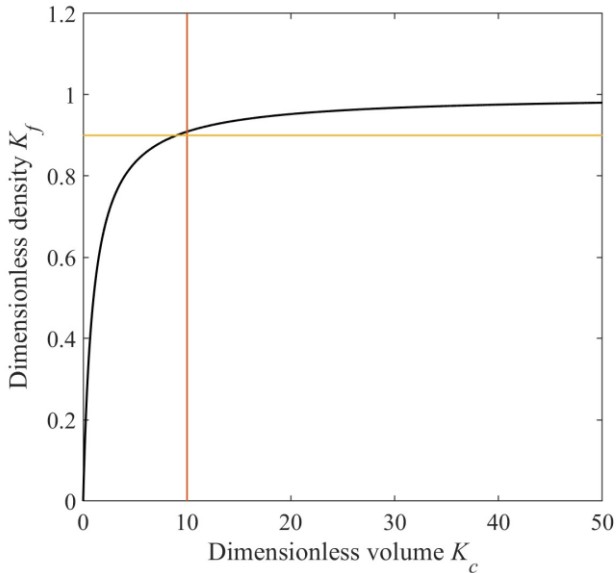


Fig. 1. Dimensionless density, $K_f$, as a function of dimensionless volume $K_c$.
Major variations in $K_f$ occur for $K_c < 10$ indicating longer gas transmission duration
with more pressure-decay data available for permeability derivation.

**3.2 Recommendation for solution selection**
The following three aspects need to be considered before selecting the
appropriate solution for permeability calculation: 1) early- or late-time
solutions; 2) error between the approximate and exact solutions; and 3) the
convenience and applicability of solutions suitable for different experiments.
We will first discuss the selection criteria for early- or late-time solutions.
Fig. 2(a) shows the exact solution of $F_s$ with their two approximate early-
and late-time solution (Table 1). Two exact solutions of $F_f$ where $K_c$ equals
to 10 or 50 are also demonstrated in Fig. 2(a). Fig. 2(b) depicts the exact
solution from $F_f$ for different $K_c$ values from 1 to 100 and their
corresponding approximate solution for Eq. (3A). The intersection point of the
solution Eq. (3B) and Eq. (3C), namely $\tau = 0.024$ in Fig. 2(a), is used for
distinguishing early- and late-time solutions.
Two notable observations can be drawn from Fig. 2(b). Firstly, the
approximate solution Eq. (3A) would only be applicable at late times when
$\tau$ is longer than 0.024. For $\tau < 0.024$, regardless of the $K_c$ value, Eq. (3C)
would be more precise than Eqs. (3A) and (3B) and return results close to the
exact solution for both $F_f$ and $F_s$. Secondly, results of Eqs. (3A) and (3B)
presented in Fig. 2(a) are similar; there difference is very small especially
for $K_c > 10$. Due to the fact that core samples from deep wells are relatively
short in length and their void volume is small (ultra-low porosity and
permeability such as in mudrocks with $k \leq 0.1$ nD), in practice, a solution for
$10 < K_c < 100$ is the most common outcome, even if the sample cell is loaded
as full as possible. Under such circumstances, the newly derived solution, Eq.
(3B), becomes practical and convenient: 1) if the $K_c$ and dimensionless time
$\tau$ have not been evaluated precisely before the GPT experiment, this solution
may fit most experimental situations; 2) this solution is suitable for calculation
as it does not need the solution from the transcendental equation of Eq. (S30)
because the denominator of $\alpha$ has been replaced by $\pi$. The data quality
control is discussed in Section 4.1.

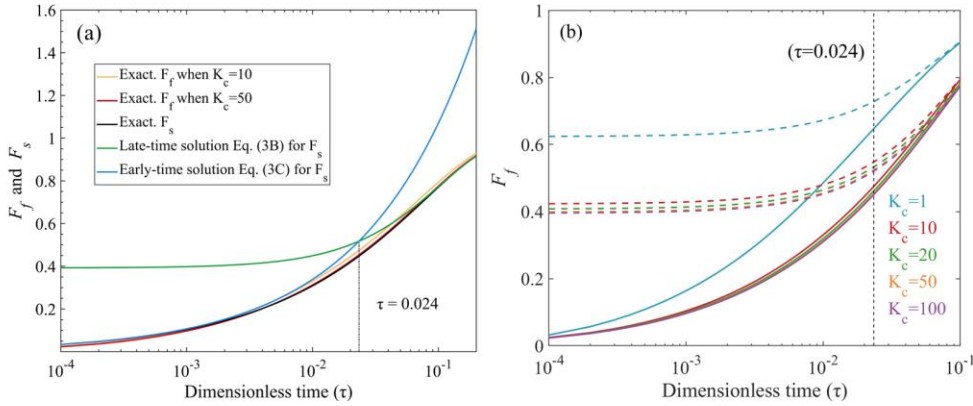


Fig. 2. Three GPT solutions with different values of $\tau$, $K_c$; the dashed lines are

approximate solutions without a series expansion in Fig. (2b) for $F_f$. Figure

modified from Cui et al. (2009).

**3.3 Applicability of the early-time solution**

A small $K_c$ value can guarantee a sufficient time for gas transfer in samples

and provide enough linear data for fitting purposes. We note that the selection

of the limited $K_c$ solution of $F_f$, and the infinity $K_c$ solution $F_s$ is controlled

by $K_c$. However, before the selection of $K_c$, the dimensionless time is the

basic parameter to be estimated as a priori before the early- or late-time

solutions are selected.

For pulse-decay methods, the early-time solution has the advantage of

capturing the anisotropic information contained in reservoir rocks (Jia et al.,

2019; Kamath, 1992). However, it suffers from the shortcoming of uncertainty

in data for initial several seconds, which as a result is not recommended for

data processing (Brace et al., 1968; Cui et al., 2009). This is due to: (1) the

Joule-Thompson effect, which causes a decrease in gas temperature from the

expansion; (2) kinetic energy loss during adiabatic expansion; and (3) collision

between molecules and the container wall. These uncertainties normally occur

in the first 10-30 sec, shown in our experiments as a fluctuating period called

"Early Stage".

However, the "Early Stage" present in pulse-decay experiments does not

mean that the early-time solution is not applicable. We demonstrate the
relationship between time and dimensionless time in Fig. 3 that a short
dimensionless time may correspond to a long testing period of hundred to
thousand seconds in experiments. This is particularly noticeable for the ultra-
low permeability samples with $k \leq 0.1$ nD and small dimensionless times
$\tau < 0.024$. This situation would only be applicable to early-time solution, but
with data available beyond the "Early Stage" and provide available data in a
long time (hundreds to thousands of seconds). For example, the early-time
solution would fit ultra-low permeability samples in 600s for 0.1 nD, and at
least 1000s for 0.01 nD shown in Fig. 3 in the region below the dark line. Then,
using Eq. (3C), the derived permeability would be closer to its exact solution
in the earlier testing time (but still after the "Early Stage"). The mudrock
samples that we tested, with results presented in Section 5.3, exhibit low
permeabilities, approximately on the order of 0.1 nD.

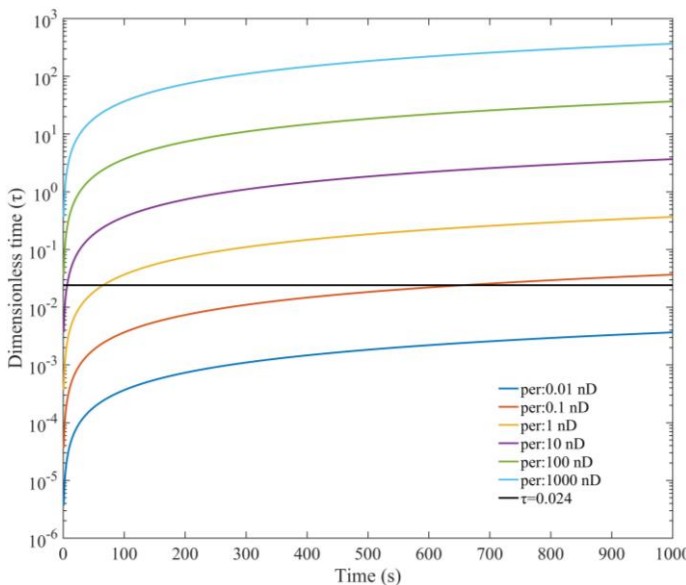

Fig. 3 Dimensionless time $\tau$ versus actual times for different permeability values

trough Eq. (S14) using He gas, sample porosity of 5%, and sample diameter of 2

mm.

**3.4 Error analyses between exact and approximate solutions**

It is unpractical to use the exact solutions with their series part to do the

permeability calculation; thus, only the approximate solutions are used and the

error difference between the exact and approximate solutions is discussed here.

The original mathematical solutions, Eqs. (S39) and (S49), are based on

series expansion. For dimensionless densities $F_f$ and $F_s$ in Eqs. (S39) and

(S49), their series expansion terms should converge. However, the rate of

convergence is closely related to the value of $\tau$. For example, from Eq. (S30),

when $\tau \geq 1$, the exponent parts of $U_s$ and $U_f$ are at least $(2n + 1)\pi^2$.

Therefore, the entire series expansion term can be omitted without being
influenced by $K_c$. In practical applications, the solutions given in Eqs. (3A)-
(3C) are approximates without series expansion. In this study, we provide the
diagrams of change in errors with dimensionless time in the presence of
adsorption (Fig. 4).
For $F_f$, the error differences between the exact and approximate solutions
are 3.5% and 0.37% for $\tau = 0.05$ and 0.1 when $K_c = 10$, respectively. When
$\tau \leq 0.024$, the error would be greater than 14.7%. Fig. 2(b) shows that $F_f$
can be approximated as $F_s$ when $K_c$ is greater than 10; the error difference
between $F_f$ and $F_s$ is quite small at this $K_c$ value (for $K_c = 10$, 6.6% is the
maximum error when $\tau = 0.01$; 4.4% when $\tau = 0.05$; and 2.9% when $\tau = 0.1$)
as shown in Fig. 4.
For $F_s$, the error difference is roughly the same as $F_f$ and equal to 3.6%
for $\tau = 0.05$ and 0.38% for $\tau = 0.1$. This verifies that newly derived Eq. (3B)
is equivalent to Eq. (3A) when $K_c$ is greater than 10. As for the evaluation
of Eq. (3C), the error difference with the exact solution will increase with
dimensionless time (5.1% for $\tau = 0.003$, 9.7% for $\tau = 0.01$, and 16% for $\tau =$

0.024).

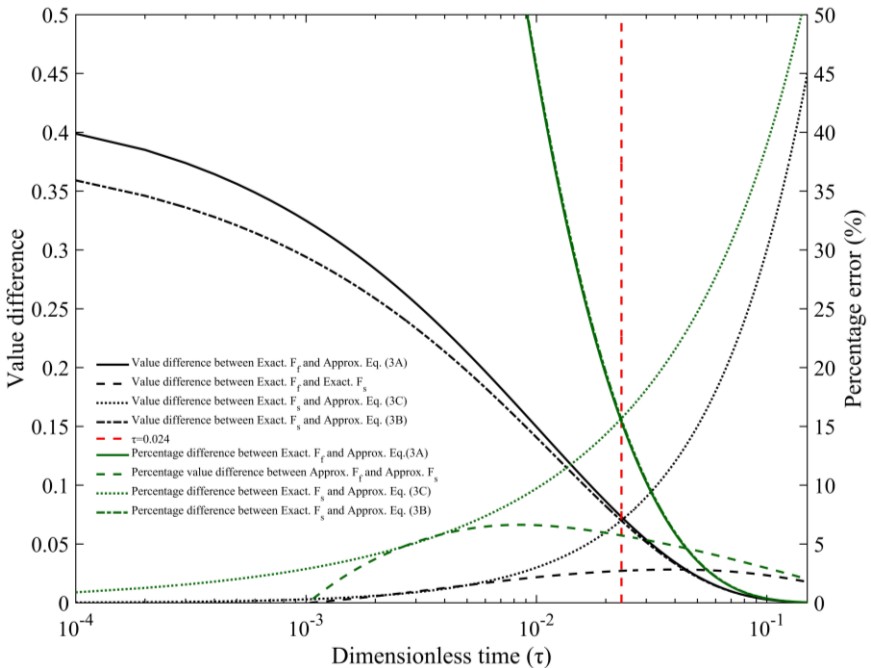

Fig. 4. Error analyses of $F_f$ and $F_s$ for their exact and approximate solutions

## 4. Influence of kinetic energy on gas transport behavior

### 4.1 Flow state of gas in granular samples

In the following, we apply the approximate solutions, Eqs. (3A-3C), to some detailed experimental data and determine permeability in several mudrock samples practically compatible with sample size, gases, and molecular dynamics analyses.

During the GPT, with the boundary conditions described in SI2, the pressure variation is captured after gas starts to permeate into the sample from the edge, and the model does not take into account the gas transport between particles or into any micro-fractures, if available. Thus, the transport that conforms to

the "unipore" model and occurs after the "Early Stage" (defined in Section 3.3)
or during the "Penetration Zone" (the area between the two vertical lines in
Fig. 5), should be used to determine the slope. Fig. S2 shows how to obtain
the permeability result using the applicable mathematical solutions (Eqs. 3A-
C). Fig. 5 shows the pressure variance with time during the experiment using
sample size from 0.34 mm to 5.18 mm for sample X-1 and sample X-2. From
Fig. 5, the time needed to reach pressure equilibrium after the initial
fluctuation stage is 20-100 sec, and the "Penetration Zone" decreases with
decreasing grain size over this time period.

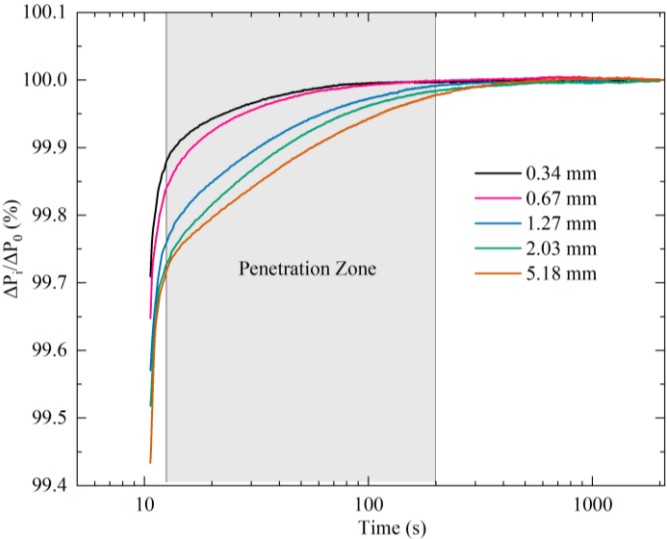


Fig. 5. Fitting region (the "Penetration Zone" in the shadowed area) for mudrock
sample X-1 with different granular sizes; the penetration zone illustrating the

pressure gradient mainly happens at 20 to 200 sec for this sample.


In fact, the "Penetration Zone", as an empirical period, is evaluated by the
pressure change over a unit of time before gas is completely transported into
the inner central part of the sample to reach the final pressure. Owing to the
sample size limitation, a decreasing pressure could cause multiple flow states
(based on the Knudsen number) to exist in the experiment. The pressure during
the GPT experiment varies between 50 and 200 psi (0.345 MPa to 1.38 MPa).
Fig. 6 shows the Knudsen number calculated from different pressure
conditions and pore diameters together with their potential flow state. Based
on Fig. 6, the flow state of gas in the GPT experiments is mainly dominated
by Fickian and transition diffusion. Essentially, the flow state change with
pressure should be strictly evaluated through the Knudsen number in Fig. 6 to
guarantee that the data in the "Penetration Zone" are always fitted with the
GPT's constitutive equation for laminar or diffusive states. This helps obtain a
linear trend for $ln(1 - F_f)$ or $F_s{}^2$ versus time for low-permeability media.
Experimentally, data from 30 to several 100 seconds are recommended for
tight rocks like shales within the GPT methodology.

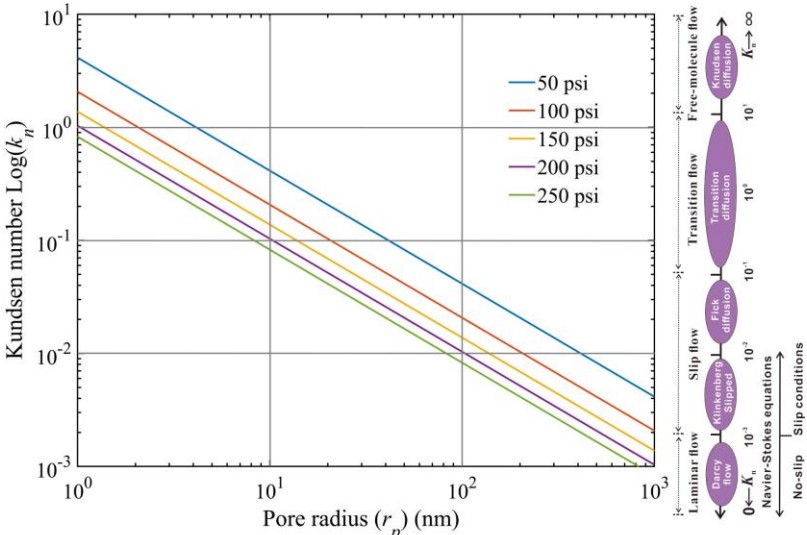


Fig. 6. Flow state of gas under diffferent testing pressures; modified from Chen &

Pfender (1983) and Roy et al. (2003) (Chen and Pfender, 1983; Roy et al., 2003).

In the GPT approach, as mentioned earlier, Eq. (S33) holds for small $K_c$

values (e.g., < 10) so that the approximately equivalent void volume in the

sample cell and sample pore volume would allow for sufficient pressure drop.

It also gives time and allows the probing gas to expand into the matrix pores

to have a valid "Penetration Zone" and to determine the permeability. Greater

values of $K_c$ would prevent the gas flow from entering into a slippage state

as the pressure difference would increase with increasing $K_c$. However, large

pressure changes would result in a turbulent flow (Fig. 6), which would cause

the flow state of gas to be no longer valid for the constitutive equation of the

GPT. Overall, the GPT solutions would be applicable to the gas permeability

measurement, based on the diffusion-like process, from laminar flow to
Fickian diffusion, after the correction of the slippage effect.
**4.2 Pressure decay behavior of four different probing gases**
We used three inert gases, including He, $N_2$, and Ar, and one sorptive gas
i.e., $CO_2$ (Busch et al., 2008), to compare the pressure drop behavior for
sample size with an average granular diameter of 0.675 mm. Results for the
mudrock sample X-2 are presented in Fig. 7. Among the three inert gases,
helium and argon required the shortest and longest time to reach pressure
equilibrium (i.e., He<$N_2$<Ar). In terms of pressure drop, argon exhibited the
most significant decrease. In a constant-temperature system, the speed (or rate)
at which gas molecules move is inversely proportional to the square root of
their molar masses. Hence, it is reasonable that helium (with the smallest
kinetic diameter of 0.21 nm) has the shortest equilibrium time. However, the
pressure drop is more critical than the time needed to reach equilibrium for the
GPT, as the equilibrium time does not differ much (basically within 10 seconds
for a given sample weight, except for the adsorptive $CO_2$). Argon may provide
a wider range of valid Penetration Zones in a short time scale for its longest
decay time except for adsorbed gas of $CO_2$; a choice of inert and economical
gas is suggested for the GPT experiments.

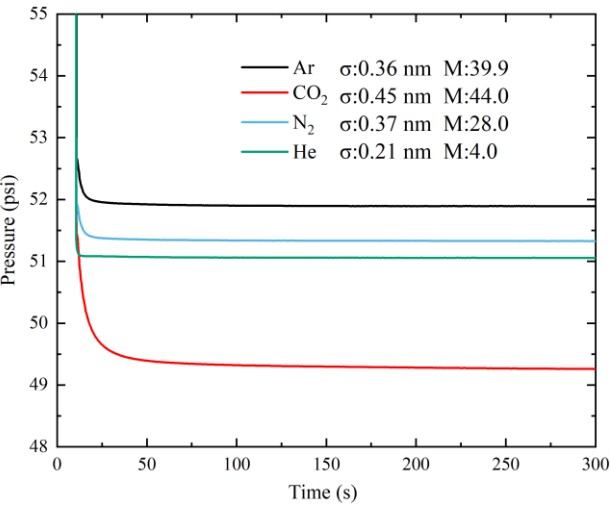


Fig. 7. Measured pressure decay curves from mudrock Sample X-2 for gases of

different molecular diameters $\sigma$ and molecular weights M (g/mol).

Fig. 7 shows that the pressure decay curve of the adsorptive gas $CO_2$ is

different from those of the inert gases used in this study. $CO_2$ has a slow

equilibrium process due to its large molar mass, and the greatest pressure drop

among the four gases due to its adsorption effect. This additional flux needs to

be taken into account to obtain an accurate transport coefficient. Accordingly,

multiple studies including laboratory experiments (Pini, 2014) and long-term

field observations (Haszeldine et al., 2006; Lu et al., 2009) were carried out to

assess the sealing efficiency of mudrocks for $CO_2$ storage. In fact, the GPT

can supply a quick and effective way to identify the adsorption behavior of

different mudrocks for both laminar-flow and diffusion states.

**4.3 Pressure decay behavior for different granular sizes**

We compared the pressure drop behavior of gas in the mudrock Sample X-1 with different granular sizes (averaged from 0.34 mm to 5.18 mm) using the same sample weight and $K_c$. Results based on the experimental data shown in Fig. 8 indicate that a larger-sized sample would provide more data to be analyzed for determining the permeability. This is because the larger the granular size, and (1) the larger the pressure drop, (2) the longer the decay time as Fig. 8 demonstrates. This is consistent with the simulated results reported by Profice et al. (2012).

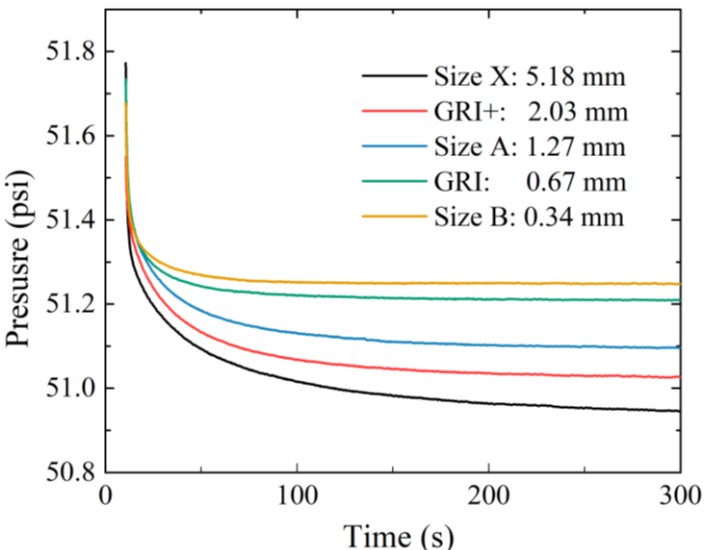

Fig. 8. Pressure decay curves measured by helium on sample X-1 with five different granular sizes. The intra-granular porosity was 5.8% independently measured by mercury intrusion porosimetry.


Table 2. Permeability results from the methods of GPT and SMP-200 for X-1.

| Granular size (mm) | SMP-200 (nD) [§] | GPT test 1 (nD) [£] | GPT test 2 (nD) [£] | Average value (nD) [£] | Fitting duration (s) | Unselected Solution (nD) [*] | Dimensionless time | Particle density (g/cm$^3$) |
|---|---|---|---|---|---|---|---|---|
| 5.18 | - | 1.17 | 1.17 | 1.17(ILT) | 50-100 | 239(IET) 1.31(LLT) | 0.023-0.027 | 2.631 |
| 2.03 | 14.2 | 0.45 | 0.41 | 0.43(LLT) | 50-100 | 11.1(IET) 0.36(ILT) | 0.026-0.028 | 2.626 |
| 1.27 | - | 0.10 | 0.10 | 0.10(ILT) | 30-60 | 20.5(IET) 0.09(ILT) | CR[*] | 2.673 |
| 0.67 | 0.65 | 0.08 | 0.04 | 0.06(LLT) | 30-60 | 1570(IET) 0.03(ILT) | CR[*] | 2.658 |
| 0.34 | - | 0.02 | - | 0.02(IET) | 30-60 | 0.00076(LLT) 0.00068(BLT) | CR[*] | 2.643 |

[§] The results are from the SMP-200 using the GRI default method.

[£] The results are from the GPT method we proposed.

[*] CR means the conflict results that the verified dimensionless time does not confirm the early- or late-time solutions using the solved permeability. For example, the verified dimensionless time would be > 0.024 using the early-time solution solved result and vice versa.

[*] represents the result which failed for the criteria of dimensionless time

As reported in Table 2, the permeability values measured by the GPT

method are one or two orders of magnitude greater than those measured by the
SMP-200 instrument. The built-in functions of SMP-200 can only be used for
two default granular sizes (500-841 μm for GRI and 1.70-2.38 mm for what
we call GRI+) to manually curve-fit the pressure decay data and determine the
permeability. The GRI method built in the SMP-200 only suggests the fitting
procedure for data processing without publicly available details of underlying
mathematics. The intra-granular permeabilities of mudrocks sample X-1 vary
from 0.02 to 1.17 nD for five different granular sizes using the GPT. With the
same pressure decay data selected from 30 to 200 sec, the permeability results
for GRI and GRI+ sample sizes from the SMP-200 fitting are 0.65 and 14.2
nD, as compared to 0.06 and 0.43 nD determined by the GPT using the same
mean granular size. Our results are consistent with those reported by Peng &
Loucks (2016) who found two to three orders of magnitude differences
between the GPT and SMP-200 methods (Peng and Loucks, 2016).
There exist several issues associated with granular samples with
diameters smaller than on average 1.27 mm. First, the testing duration is short,
and second, there would not be sufficient pressure variation analyzed in Fig.
8. Both may cause significant uncertainties in the permeability calculation and,
therefore, make samples with diameters smaller than 1.27 mm unfavorable for
the GPT method, particularly extra-tight (sub-nD levels) samples, as there is
almost no laminar or diffusion flow state to be captured. The greater pressure
drop for larger-sized granular samples would result in greater pressure
variation and wider data region compared to smaller granular sizes (see Figs.
6 and 9). Although samples of large granular sizes may potentially contain
micro-fractures, which complicate the determination of true matrix
permeability (Heller et al., 2014), the versatile GPT method can still provide
size-dependent permeabilities for a wide range of samples (e.g., from sub-mm
to 10 cm diameter full-size cores) (Ghanbarian, 2022a, b). Besides, the surface
roughness of large grains may also complicate the determination of
permeability, which need to pay attention to (Devegowda, 2015; Rasmuson,
1985; Ruthven and Loughlin, 1971). Overall, our results demonstrated that
sample diameters larger 2 mm are recommended for the GPT to determine the
nD permeability of tight mudrocks, while smaller sample sizes may produce
uncertain results.
**4.4 Practical recommendations for the holistic GPT**

Here, we evaluate the potential approximate solution for tight rock

samples using frequently applied experimental settings by considering the
critical parameters, such as sample mass, porosity, and estimated permeability
(as compiled in Fig. 9 showing the dimensionless time versus porosity). Based
on the results presented in Figs. 3 and 6, only $t < 200s$ is dominant and critical
for the analyses of dimensionless time and penetration zone. Thus, we take
200s and use helium to calculate the dimensionless time. Another critical
parameter to assure enough decay data is the sample diameter greater than 2
mm. Thus, we only show the dimensionless time versus porosity for sample
diameter greater than the criteria of 2 mm.

Fig. 9 demonstrates that the sample permeability has dominant control on

the early- or late-solution selection, and we decipher a concise criterial for
three solutions selection. We classify the dimensionless time versus porosity
relationship into three cases. Firstly, among the curves shown in Fig. 9, only
that corresponding to $k = 0.1$ nD and sample diameter of 2 mm stays below
the dashed line representing $\tau = 0.024$. Therefore, the early time solution is
appropriate for tight samples with permeabilities less than 0.1 nD (as shown
in the analyses of Section 4.3, which also conforms to the situation of the
molecular sieve sample that we tested in SI3). Secondly, for permeabilities
greater than 10 nD (the curve is above the line of $\tau = 0.024$), the new derived
late-time solution, Eq. (3B), is recommended as it is more convenient for
mathematical calculation without the consideration of transcendental
functions. The reason is that the sample cell can be filled as much as possible
(~90% of the volume) with samples and solid objects. However, as the tight
rock hardly presents a large value of porosity, the small $K_c$ value is difficult
to be achieved with an inconsequential influence between Eq. (3B) and Eq.
(3A). Lastly, in the case of permeability around 1 nD, the value of porosity
would be critical in the selection of the early- or late-time solutions, as shown
in Fig. 9.

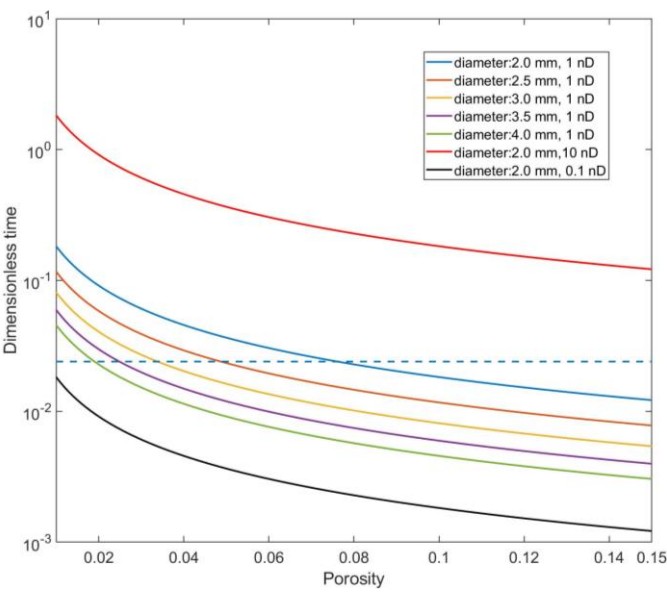


Fig. 9. Holistic GPT to explore the appropriate solution based on diameter,

permeability, and porosity of samples. The legend shows the diameter of granular

sample and permeability, along with a dashed line for dimensionless time of 0.024,

while regions above and below this value fit for the late- and early-time solutions,

respectively.

## 5. Conclusions

In the present work, we solved fluid flow state equation in granular porous

media and provided three exact mathematical solutions along with their

approximate ones for practical applications of low permeability measurements.

The mathematical solutions for the transport coefficient in the GPT were

derived for a spherical coordinate system, applicable from laminar flow to

slippage-corrected Fickian diffusion. Of the three derived solutions, one is

valid during early times when the gas storage capacity $K_c$ approaches infinity,

while the other two are late-time solutions to be valid when $K_c$ is either small
or tends towards infinity. We evaluated the derived solutions for a systematic
measurement of extra-low permeabilities in granular media and crushed rocks
using experimental methodologies with the data processing procedures. We
determined the error for each solution by comparing with the exact solutions
presented in the SI. The applicable conditions for such solutions of the GPT
were investigated, and we provided the selection strategies for three
approximate solutions based the range of sample permeability. In addition, a
detailed utilization of GTP was given to build up the confidence in the GPT
method through the molecular sieve sample, as it enables a rapid permeability
test for ultra-tight rock samples in just tens to hundreds of seconds, with good
repeatability.
**Data availability.** This work did not use any data from previously published
sources, and our experimental data & processing codes of MATLAB are
available at https://doi.org/10.18738/T8/YZJS7Y, managed by Mavs
Dataverse of the University of Texas at Arlington.
**Supplement.** An early-version preprint of this work appears as DOI:
10.1002/essoar.10506690.2 (Zhang et al., 2021).
**Author contributions.** TZ and QHH planned and designed the research,
performed the analyses, and wrote the paper with contributions from all co-
authors. BG, DK, and ZM participated in the research and edited the paper.
**Competing interest.** We declare that we do not have any commercial or

associative interest that represents a conflict of interest in connection with the work submitted.

**Acknowledgments.** Financial assistance for this work was provided by the National Natural Science Foundation of China (41830431; 41821002), PetroChina International Cooperation Project (2023DQ0422), Shandong Provincial Major Type Grant for Research and Development from the Department of Science & Technology of Shandong Province (2020ZLYS08), Maverick Science Graduate Research Fellowship for 2022-2023, the AAPG and the West Texas Geological Foundation Adams Scholarships, and Kansas State University through faculty start-up funds to BG. We extend our deepest appreciation to handling editor Dr. Monica Riva for her diligent assistance, and to the two anonymous referees for their insightful comments on this paper.

## Nomenclature

$B_{ij}$ Correction parameter for viscosity, constant

$c_t$ Fluid compressibility, Pa$^{-1}$

$F_f$ Uptake rate of gas outside the sample, dimensionless

$F_s$ Uptake rate in the sample, dimensionless

$f_1$ Intercept of Eq. (S40), constant

$K_a$ Apparent transport coefficient defined as Eq. (S9), m$^2$/s

$K_c$ Ratio of gas storage capacity of the total void volume of the system to the pore (including adsorptive and non-adsorptive transport) volume of the sample, fraction

$K_f$ Initial density state of the system, fraction

$k$ Permeability, m$^2$

$k_s$ Permeability defined as Eq. (S8), m$^2$/(pa·s)

$L$ Coefficient, unit for certain physical transport phenomenon

$M$ Molar mass, kg/kmol

$M_m$ Molar mass of the mixed gas, kg/kmol

$M_{i,j}$ Molar mass for gas i or j, kg/kmol

$M_s$ Total mass of sample, kg

$N$ Particle number, constant

$p$ Pressure, Pa

$p_{cm}$ Virtual critical pressure of mixed gas, Pa

$p_p$ Pseudo-pressure from Eq. (S1), Pa/s

$R_a$ Particle diameter of sample, m

$R$ Universal gas constant, 8.314 J/(mol·K)

$r$ Diameter of sample, m

$s_1$ Slope of Eq. (S40), constant

$s_2$ Slope of function $Ln(1 - F_s)$, constant
$s_3$ Slope of function $F_s{}^2$, constant
$T$ Temperature, K
$T_{cm}$ Virtual critical temperature for mixed gas, K
$t$ Time, s
$U_f$ Dimensionless density of gas outside the sample, dimensionless
$U_s$ Dimensionless density in grain, dimensionless
$U_\infty$ Maximum density defined as Eq. (S37), dimensionless
$V_1$ Cell volume in upstream of pulse-decay method, $m^3$
$V_2$ Cell volume in downstream of pulse-decay method, $m^3$
$V_b$ Bulk volume of sample, $m^3$
$V_c$ Total system void volume except for sample bulk volume, $m^3$
$\overline{v}$ Dacian velocity in pore volume of porous media, m/s
$X$ Pressure force, Pa
$y_{i,j}$ Molar fraction for gas i or j, fraction
$z$ Gas deviation (compressibility) factor, constant
**Greek Letters:**
$\alpha_n$ The nth root of Eq. (S30), constant
$\mu$ Dynamic viscosity, pa·s or N·s/m$^2$
$\mu_{i,j}$ Dynamic viscosity for gas i or j, pa·s or N·s/m$^2$
$\mu_{mix}$ Dynamic viscosity of mixture gas, pa s or N s/m$^2$
$\mu_p$ Correction term for the viscosity with pressure, pa s or N s/m$^2$
$\xi$ Dimensionless radius of sample, dimensionless
$\rho$ Density of fluid, kg/m$^3$
$\rho_0$ Average gas density on the periphery of sample, kg/m$^3$
$\rho_1$ Gas density in reference cell, kg/m$^3$
$\rho_2$ Gas density in sample cell, kg/m$^3$
$\rho_b$ Average bulk density for each particle, kg/m$^3$
$\rho_f$ Density of gas changing with time outside sample, kg·m$^{-3}$·s$^{-1}$
$\rho_{f\infty}$ Maximum value of $\rho_f$ defined as Eq. (S38), kg·m$^{-3}$·s$^{-1}$
$\rho_p$ Pseudo-density from Eq. (S1), kg·m$^{-3}$·s$^{-1}$
$\rho_s$ Density of gas changing with time in sample, kg·m$^{-3}$·s$^{-1}$
$\rho_{ps}$ Pseudo-density of gas changing with time in sample, kg·m$^{-3}$·s$^{-1}$
$\rho_{pf}$ Pseudo-density of gas changing with time outside sample, kg·m$^{-3}$·s$^{-1}$
$\rho_{p2}$ Initial pseudo-density of gas in sample, kg·m$^{-3}$·s$^{-1}$
$\rho_{p0}$ Average pseudo-density of gas on sample periphery, kg·m$^{-3}$·s$^{-1}$
$\rho_{rm}$ Relative density to the mixed gas, kg·m$^{-3}$·s$^{-1}$
$\rho_{sav}$ Average value of $\rho_{sr}$ defined as Eq. (S47), kg·m$^{-3}$·s$^{-1}$
$\rho_{sr}$ Average value of pseudo-density of sample changing with diameter,
619        kg·m$^{-3}$·s$^{-1}$

$\rho_{s\infty}$ Maximum value of $\rho_{sr}$ defined as Eq. (S46), kg·m$^{-3}$·s$^{-1}$
$\tau$ Dimensionless time, dimensionless
$\phi$ Sample porosity, fraction
$\phi_f$ Total porosity ($\phi_f = \phi_a + \phi_b$) occupied by both free and adsorptive
fluids, fraction

**Supporting Information (SI)**

**SI1. Consideration of Non-linearity of Gas and Solutions for a Mixed Gas State**

For gas flow, we can use a pseudo-pressure variable to linearize Eq. (2A) as $\mu$ and $c_t$ are functions of pressure. The pseudo-pressure $p_p$ is defined as (Haskett et al., 1988)

$$p_p = 2 \int_{p_0}^{p} \frac{p}{\mu z} d\,p \qquad (S1)$$

By combining Eq. (S1) with the ideal gas law, the pseudo-density may be expressed as

$$\rho_p = \frac{pM}{RT} = \frac{p^2 M}{\mu z RT} \qquad (S2)$$

Because viscosity and compressibility do not change significantly (less than 0.7%) between 200 psi and atmospheric pressures, Eq. (S2) can be simplified to

$$\rho_p = \frac{p^2 M}{RT} \qquad (S3)$$

Thus, the density change is replaced by the pseudo-density for a precise calibration by using pressure squared.

During the GPT experiment, different gases in the reference and sample cells may complicate the hydrodynamic equilibrium of gas, and consequently the expression of transport phenomena, as the viscosity and gas compressibility are in a mixed state. Therefore, during the GPT experiment

when a different gas exists between the reference and sample cells a, a mixed
viscosity should be used after the gas in reference cell is released into the
sample cell. The viscosity of mixture $\mu_{mix}$ under pressure in Eqs. (3A)-(3C)
can be calculated from (Brokaw, 1968; Sutherland, 1895)
$$\mu_{mix} = \sum \frac{\mu_i}{1+\frac{1}{y_i}(\sum_{\substack{j=1\\j\neq i}}^{n} B_{ij}y_j)} + \mu_p \tag{S4}$$

$B_{ij}$ is a correction parameter independent of gas composition and can be
expressed as
$$B_{ij} = \frac{[1+(\frac{\mu_i}{\mu_j})^{0.5}(\frac{M_j}{M_i})^{0.5}]^2}{2\sqrt{2}(1+\frac{M_j}{M_i})^{0.5}} \tag{S5}$$

in which $\mu_p$ is the correction term for the viscosity variation as its changes
with pressure and given by
$$\mu_p = 1.1 \times 10^{-8}(e^{1.439\rho_{rm}} - e^{-1.111\rho_{rm}^{1.858}}) \times M_m^{0.5} \cdot \frac{P_{cm}^{2/3}}{T_{cm}^{1/6}} \tag{S6}$$

**SI2. Gas Transport in GPT**
From Eq. (2A), the transport of gas in the GPT with the "unipore" model
under a small pressure gradient in a spherical coordinate system with laminar
flow is based on the Darcy-type relation. Because the transfer rate of the fluid
is proportional to the concentration gradient, this process can be expressed as:
$$\frac{\partial \rho_p}{\partial t} = \frac{k}{c_t \phi_f \mu}(\frac{2}{r}\frac{\partial \rho_p}{\partial r} + \frac{\partial^2 \rho_p}{\partial r^2}) \tag{S7}$$

We set
$$k_s = \frac{k}{\mu} \tag{S8}$$
$$K_a = \frac{k_s}{c_t \phi_f} \tag{S9}$$
Then, Eq. (S7) becomes:
$$\frac{\partial \rho_p}{\partial t} = K_a \left( \frac{2}{r} \frac{\partial \rho_p}{\partial r} + \frac{\partial^2 \rho_p}{\partial r^2} \right) \text{ or } \frac{\partial}{\partial t}(\rho_p r) = K_a \frac{\partial^2}{\partial r^2}(\rho_p r) \tag{S10}$$
We next introduce the following dimensionless variables:
$$U_s = \frac{r}{R} \frac{(\rho_{ps} - \rho_{p2})}{(\rho_{p0} - \rho_{p2})} \tag{S11}$$
$$U_f = \frac{\rho_{pf} - \rho_{p2}}{\rho_{p0} - \rho_{p2}} \tag{S12}$$
$$\xi = \frac{r}{R} \tag{S13}$$
$$\tau = \frac{K_a t}{R^2} \tag{S14}$$
where $\rho_1$ and $\rho_2$ are the gas density in the reference and sample cells, and
$\rho_0$ is the gas density outside the connected pore volume (the gas has flowed
from the reference into sample cells but not into samples), and $\rho_0$ is given by
$$\rho_0 = \frac{V_1 \rho_1 + (V_2 - V_b)\rho_2}{V_c} \tag{S15}$$
where $V_1$ is the reference cell volume, $V_2$ is the sample cell volume, $V_b$ is
the bulk volume of the sample, $V_c$ is the total void volume of the system
minus $V_b$ where $V_c = V_1 + V_2 - V_b$.
If the bulk density of the sample is $\rho_b$ and the total mass of the sample is
$M_s$, then the total number of sample particles $N$ is:

$$N = \frac{M_s}{\frac{4}{3}\pi R_a{}^3 \rho_b} \tag{S16}$$

Based on Darcy's law, the gas flow into a sample $Q$ is:

$$Q = -4\pi R^2 (k_s \frac{\partial p}{\partial r}) N = -\frac{3}{R} \frac{M_s}{\rho_b} k_s \frac{\partial p}{\partial r} \tag{S17}$$

According to mass conservation and in combination with Eq. (S17), for

$t > 0$ and $r = R_a$, we have

$$-\frac{3}{R} V_b K_a c_t \phi_f \frac{\partial p}{\partial r} \rho_s = V_c \frac{\partial \rho_f}{\partial t} \tag{S18}$$

Substituting Eq. (1C) into Eq. (S18), the boundary condition of Eq. (S10),

for $\xi = 1$, is:

$$-\frac{3}{R} V_b K_a \phi_f \frac{\partial \rho_s}{\partial r} = V_c \frac{\partial \rho_f}{\partial t} \tag{S19}$$

Substituting dimensionless variables into Eq. (S10) yields:

$$\frac{\partial U_s}{\partial \tau} = \frac{\partial^2 U_s}{\partial \xi^2} \tag{S20}$$

By defining parameter $K_c$ as:

$$K_c = \frac{V_c}{V_b \phi_f} \tag{S21}$$

the boundary condition of Eq. (S19) becomes:

$$\frac{\partial U_f}{\partial \tau} = -\frac{3}{K_c} \left( \frac{\partial U_s}{\partial \xi} - \frac{U_s}{\xi} \right) \tag{S22}$$

From Eq. (S21), $K_c$ represents the ratio of gas storage capacity of the total

void volume of system to the pore volume (including both adsorption and non-

adsorption volume) of sample.

The initial condition of Eq. (S20), for $\tau = 0$, is:

when $0 \leq \xi < 1, U_s = 0$                              (S23)

For $\tau > 0$:

$\xi = 0, U_s = 0$                                      (S24)

$\xi = 1, U_s = U_f = 1$                                  (S25)

$\frac{\partial U_s}{\partial \tau} = \frac{\partial^2 U_s}{\partial \xi^2}, 0<\xi<1$              (S26)

Replacing the Heaviside operator $p = \partial/\partial\tau$ as $p = -s^2$, Eq. (S20) and
Eq. (S22) then become:

$\left.\frac{\partial^2 U_s}{\partial \xi^2} + s^2 U_s = 0\right|_{U_s=0,\xi=0}$              (S27)

$\left.\alpha^2(U_s - 1) = \frac{3}{K_c}(\frac{\partial U_s}{\partial \xi} - \frac{U_s}{\xi})\right|_{\xi=1}$              (S28)

For these first- and second-order ordinary differential equations, we can
solve Eqs. (S27) and (S28) as:

$U_s = \frac{\alpha^2 \sin \alpha\xi}{\frac{3}{K_c}(\sin \alpha - \alpha \cos \alpha)+\alpha^2 \sin \alpha}$              (S29)

In Eq. (S29), $\alpha_n$ are the roots of Eq. (S30):

$\tan \alpha = \frac{3\alpha}{3+\alpha^2 K_c}$                              (S30)

Defining the numerator and denominator of Eq. (S29) as functions
$f(\alpha)$ and $F(\alpha)$, $U_s$ can be expressed as:

$U_s = \underset{\alpha\to0}{F}\frac{f(\alpha)}{F(\alpha)} + 2\sum_{n=1}^{\infty}\frac{f(\alpha_n)}{\alpha_n F'(\alpha_n)}e^{-\alpha_n^2\tau}$              (S31)


**SI2-1: Solution for the Limited $K_c$ Value**
Under the condition of limited $K_c$ value, Eq. (S20) is solved with the
boundary condition of $0 < \xi < 1$ at time $t$, and the gas state on the grain
surface is initially at equilibrium with the gas outside. Using the Laplace
transform, Eq. (S31) is given as (the Laplace transform part can be found in
APPENDIX V of Carslaw & Jaeger, 1959) (Brokaw, 1968; Sutherland, 1895):
$$U_s = \frac{\xi K_c}{K_c+1} + 6\sum_{n=1}^{\infty} \frac{\sin \xi \alpha_n}{\sin \alpha_n} \frac{K_c e^{-\alpha_n^2 \tau}}{9(K_c+1)+\alpha_n^2 K_c^2} \qquad (S32)$$

As the pressure transducer detects the pressure in the reference cell, with
the boundary condition $U_f = U_s|_{\xi=1}$, we can calculate $U_f$ as:
$$U_f = \frac{K_c}{1+K_c} + 6\sum_{n=1}^{\infty} \frac{K_c e^{-\alpha_n^2 \tau}}{9(K_c+1)+\alpha_n^2 K_c^2} \qquad (S33)$$

For a convenient expression of $\alpha_n$ through logarithmic equation, Eq. (S33)
can be transformed as:
$$(1 - U_f)(1 + K_c) = 1 - 6\sum_{n=1}^{\infty} \frac{K_c(1+K_c)e^{-\alpha_n^2 \tau}}{9(K_c+1)+\alpha_n^2 K_c^2} \qquad (S34)$$

The left side of Eq. (S34) clearly has a physical meaning for the state of
gas transport outside the sample, and we define $(1 - U_f)(1 + K_c)$ as $F_f$,
which is less than, but infinitely close to, 1. Parameter $F_f$ represents (1) the
fraction of final gas transfer of $V_c$ which has taken place by time $t$, which can
be interpreted as the net change in the density of gas at time $t$ to time infinity
as Eq. (S35), or (2) as the fractional approach of the gas density to its steady-
state in terms of dimensionless variables as Eq. (S36).

$$F_f = \frac{\rho_{p0} - \rho_{pf}}{\rho_{p0} - \rho_{f\infty}} \text{ or} \tag{S35}$$


$$F_f = \frac{1 - U_f}{1 - U_\infty} = \frac{\rho_{p0} - \rho_{pf}}{\rho_{p0} - \rho_{p2}}(1 + K_c) \tag{S36}$$


where for $\tau \to \infty$, the result of $U_f$ and $\rho_{f\infty}$ would tend to be the limiting
value:

$$U_\infty = U_s = U_f \xi = \left. \frac{\xi K_c}{1 + K_c} \right|_{\xi = 1} \tag{S37}$$


$$\rho_{f\infty} = \frac{V_1\rho_1 + (V_2 - V_s)\rho_2}{V_1 + V_2 - V_s} = \frac{K_c}{1 + K_c}(\rho_{p0} - \rho_{p2}) + \rho_{p2} \tag{S38}$$


Thus, Eq. (S34) can be expressed as:

$$F_f = 1 - 6\sum_{n=1}^{\infty} \frac{K_c(1 + K_c)e^{-\alpha n^2 \tau}}{9(K_c + 1) + \alpha_n^2 K_c^2} \tag{S39}$$


For calculating the permeability, Eq. (S39) can be linearized as a function
of time as there are no variables other than the exponential part:

$$ln(1 - F_f) = f_1 - s_1 t \tag{S40}$$


where $f_1$ is the intercept for the y-axis of function (S40):

$$f_1 = ln\left[\frac{6K_c(1 + K_c)}{9(1 + K_c) + \alpha_1^2 K_c^2}\right] \tag{S41}$$


The slope $s_1$ can be captured by the fitted line of the linear segment, and
$\alpha_1$ is the first solution of Eq. (S30):

$$s_1 = \frac{\alpha_1^2 K_a}{R_a^2} \tag{S42}$$


Thus, the permeability can be calculated as:

$$k = \frac{R_a^2 \mu c_t \phi_f s_1}{\alpha_1^2} \tag{S43}$$


### SI2-2: Solution for $K_c$ Goes to Infinity

When $V_c$ has an infinite volume compared to the void volume in a sample, which means that the density of gas in $V_c$ would be kept at $\rho_{p0}$, and $\alpha$ would approach $n\pi$ in Eq. (S30), then Eq. (S32) can be transformed as:

$$U_s = \xi + \frac{2}{\pi}\sum_{n=1}^{\infty}(-1)^n \frac{\sin n\pi\xi}{n} e^{-(n\pi)^2\tau} \qquad (S44)$$

In this situation, $U_f = 1$, and as the gas density would be maintained at the initial state at $\rho_{p0}$, it would be a familiar case in diffusion kinetics problems with the uptake rate of $F_f$ to be expressed as $F_s$ in $V_b$ (Barrer, 1941):

$$F_s = \frac{\rho_{sav}}{\rho_{s\infty}} \qquad (S45)$$

where $\rho_{sav}$ is the average value of $\rho_{sr}$ in the grain, and $\rho_{s\infty}$ is the maximum value of $\rho_{sr}$:

$$\rho_{sr} = \rho_{ps} - \rho_{p2}, \ \rho_{s\infty} = \rho_{p0} - \rho_{p2} \qquad (S46)$$

The value of $\rho_{sr}$ in the grain is:

$$\rho_{sav} = \frac{3}{R^3}\int_0^R \rho_{sr} r^2 \, dr \qquad (S47)$$

Then $F_s$ becomes:

$$F_s = \frac{3}{R^3}\int_0^R \frac{U_s}{\xi} r^2 \, dr \qquad (S48)$$

Substituting Eq. (S44) into Eq. (S48), we can calculate:

$$F_s = 1 - \frac{6}{\pi^2} \sum_{n=1}^{\infty} \frac{e^{-(n\pi)^2\tau}}{n^2} \qquad (S49)$$

Similar to Eq. (S39), Eq. (S49) can also be linearized to calculate the permeability in $\tau$ from the fitted slope. For $\tau \geq 0.08$, Eq. (S49) can be reduced as:

$$F_s = 1 - \frac{6}{\pi^2} e^{-\pi^2\tau} \qquad (S50)$$

When $t$ is small enough (for $\tau \leq 0.002$), Eq.(S49) can be transformed into Eq. (S51).

$$F_s = 6\sqrt{\frac{\tau}{\pi}} \qquad (S51)$$

As $F_s$ is a special solution of $F_f$ with the case of $K_c$ goes to infinity, we can arrive at:

$$F_s = F_f = (1 - U_f)(1 + K_c) \qquad (S52)$$

For testing the ultra-low permeability rocks using granular samples when $K_c$ goes to infinity, Eq. (S50) and Eq. (S51) can be selected using different $\tau$ values.

From the fitted slope $s_2$ of function $ln(1 - F_s)$ from Eq. (S50), we can then derive the permeability:

$$k = \frac{R_a{}^2 \mu c_t \phi_f s_2}{\pi^2} \qquad (S53)$$

The results of Eq. (S53) are very similar to Eq. (S43) as the first solution for Eq. (S30) is very close to $\pi$.

From the fitted slope $s_3$ of function $F_s{}^2$ from Eq. (S51), we can derive
the permeability:

$$k = \frac{\pi R_a{}^2 \mu c_t \phi_f s_3}{36}$$     (S54)


**SI3. A Case of Data Processing for GPT**
We show here an illustration of the data processing procedure for the GPT
with a molecular sieve sample (https://www.acsmaterial.com/molecular-
sieves-5a.html). This material consists of grains of 2 mm in Diameter with a
porosity of 26.28%, and a uniform pore-throat size of 5Å in Diameter, with a
particle density of 2.96 g/cm$^3$. For a 45 g sample, the $K_c$ value is 19.4 from
Eq. (S21), and therefore 4.9% of the density ratio $(1 - K_f)$ is available for
mass transfer from Eq. (1G).
The experimental data were captured under a strict temperature control and
unitary-gas environment, along with a precise measurement of barometric
pressure. The experiment was run twice, and after the data were collected, 1)
we made a rough evaluation of the "Penetration Zone" of this sample based on
Figs. 5-6. For this molecular sieve sample, the "Penetration Zone" is shown in
Fig. S1, and the mass transfer in unit time more conforming to a linear state
(shown as Fig. 5) over a large time range, especially at 100-300s; 2) data in
the selected range (100-300s) were fitted respectively for the slope from Fig.

S2, then slopes were compiled in Table SI3-1; 3) permeabilities were calculated using the slope of the fitted curve, and all results for LLT, ILT and IET are also shown in Table SI3-1; 4) the results were checked with their dimensionless times to verify whether the early- or late-time solutions were used correctly. Table SI3-1 clearly shows that the results of IET should be selected for this sample, as the dimensionless time is less than 0.024. Note that the data fluctuation shown here was from a high resolution ($\pm$0.1% for 250 psi) pressure sensor without undergoing a smoothing process; meanwhile, for data in the 100-200, 200-300, and 300-400 seconds of experimental duration, 100, 200, and 300 seconds respectively were used to calculate the dimensionless times for the results in Table SI3-1.

In addition, the validity of the permeability obtained needs to be verified by using the time interval employed in data fitting and the calculated permeability results to calculate the $\tau$ (Table SI3-1). If the dimensionless time is less than 0.024 (as occurred for the case of molecular sieve), the IET solution is selected; if the dimensionless time is greater than 0.024 and $K_c$ is greater than 10, the ILT solution is used; if $\tau$ is greater than 0.024 and $K_c$ is less than 10, then the LLT solution is employed. However, for sample sizes smaller than 1.27 mm, Conflicting Results (described in Table 1) occur, and results from this situation are not recommended due to poor data quality.


Table SI3-1. Permeability results of molecular sieve from LLT, IET and ILT

| Fitting range (s) | LLT ($m^2$) | $\tau$ - LLT | IET($m^2$) | $\tau$ - IET | ILT ($m^2$) | $\tau$ -ILT | Slope-LLT | Slope-IET | Slope-ILT |
|---|---|---|---|---|---|---|---|---|---|
| 100-200 | 5.60E-22 | 0.004 | 1.02E-21 | 0.007 | 5.00E-22 | 0.003 | 0.0004 | 0.0007 | 0.0004 |
| 200-300 | 4.20E-22 | 0.006 | 5.81E-22 | 0.008 | 3.75E-22 | 0.005 | 0.0003 | 0.0004 | 0.0003 |
| 300-400 | 2.80E-22 | 0.006 | 4.36E-22 | 0.009 | 2.50E-22 | 0.005 | 0.0002 | 0.0003 | 0.0002 |

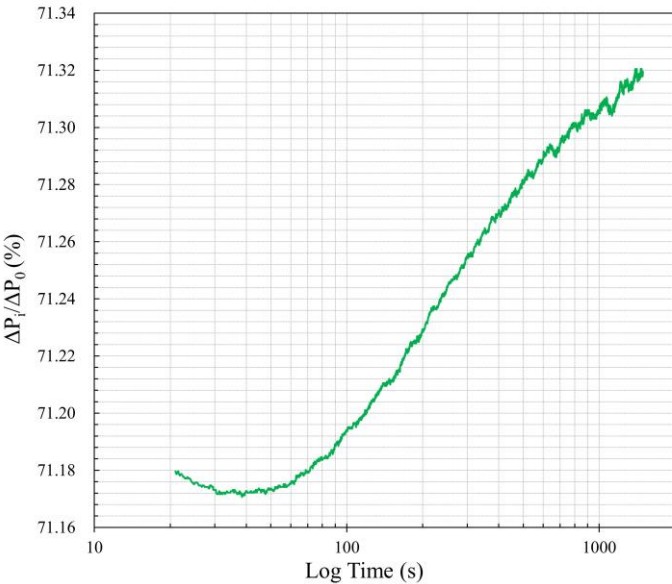


Fig. S1. Unit pressure change varying with experimental time.


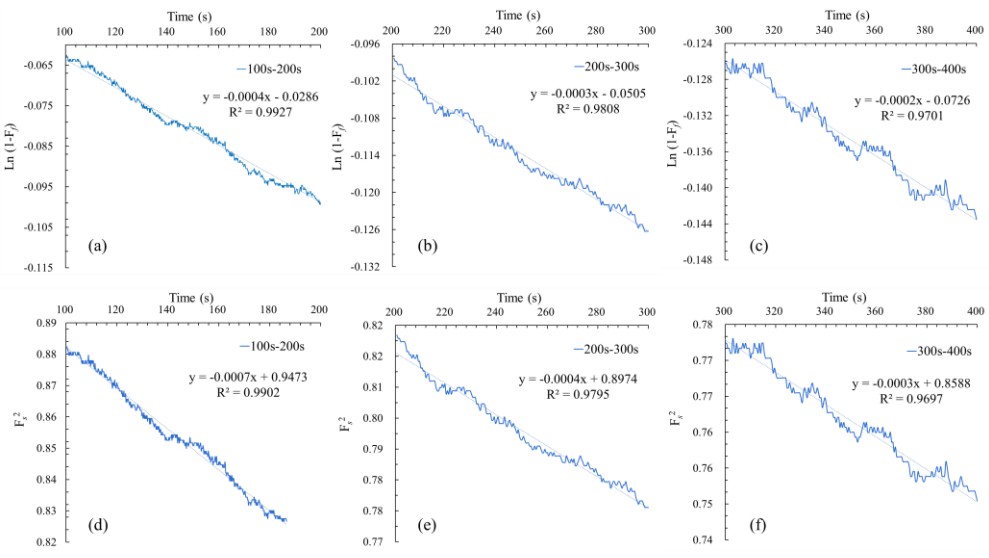


Fig. S2. Fitted slopes for each solution; (a) to (c) are results of LLT and ILT, while

(d) to (f) of IET.

**SI4. Equipment and samples**
The experimental setup in the GPT presented in this study is based on the
GRI-95/0496 protocols (Guidry et al., 1996) and the SMP-200 guidelines from
Core Laboratories with the gas expansion approach (shown in Fig. S3). In this
work, gases (He, Ar, $N_2$, or $CO_2$) with different molecular sizes and sorption
capacities were tested using two shale core samples (X1, X2) from an oil-
producing lacustrine formation in the Songliao Basin, China. X1 is used for
sample size study where X2 used for experiment with different gas. Also, we
used the molecular sieve to exhibit the practical utilization of the GPT method
in SI3. We gently crushed the intact samples with mortar and ground to
different granular sizes from 0.34 mm to 5.18 mm through a stack of sieves
(named here as Size X: 8 mm to #8 mesh; GRI+: #8-#12 mesh; Size A: #12-
#20 mesh; GRI: #20-#35 mesh; Size B: #35-#80 mesh).

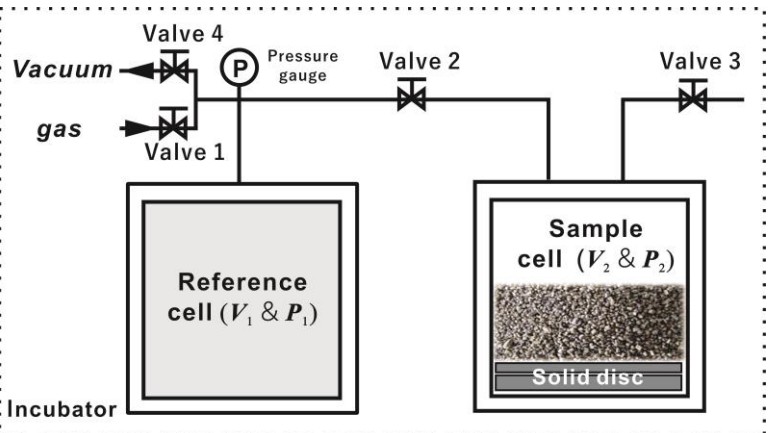


Fig. S3. Scheme of the GPT experiment for granular samples with all the cells and

supplies placed inside an incubator for temperature control.

After loading each sample, related accessories (e.g., solid discs or balls for
volume control; and hence porosity, sample mass, and solution-related) were
placed below samples inside the cell (Fig. S3). Next, valves 1 and 3 were
closed, then valves 2 and 4 were opened for air evacuation. Using a precise
pressure gauge connected to the reference cell shown in Fig. S3 we monitored
changes in the pressure. The evacuation time typically lasted at least 15-30
min, and then the system was allowed to stabilize for another 15 min. As the
moisture content of the samples significantly influences the final vacuum, the
samples were placed into the sample cell immediately after removal from the
drying oven set at 60°C for two days and cooling in a low-humidity desiccator.
The experiments were conducted at the temperature of 35°C by placing the
SMP-200 inside an incubator equipped with a high precision temperature-
humidity sensor to monitor changes. This is to ensure that the system
temperature was always stable (0.05°C over at least 45 mins of experimental
duration). For temperature monitoring, after evacuation, we closed valves 3
and 4 followed by opening valves 1 and 2 (shown in Fig. S3) and monitoring
the heat convection and conduction in the system with the pressure gauge.
Normally, the sample was placed inside the sample cell in less than 30 sec
after opening the incubator and remained at least 45 min for the gas pressure
to stabilize before the pressure decay test. After the pressure was stabilized
(0.005 psi for an experimental pulse pressure of 200 psi), it was deemed that
there was no appreciable additional flow due to temperature variation in the
system, as indicated by the rebound of the pressure decay curve. After reaching
a unitary gas condition and stable temperature in the GPT experiment, valves
2 and 4 were closed, and the reference cell was filled with the probing gas
(mostly non-reactive helium) at 200 psi. Valve 2 was then opened to release
the pressure in the reference cell into the void volume in the sample cell, and
the pressure decay for both reference and sample cells were recorded over time.
**SI5. Experimental conditions**
We performed leakage tests by measuring the pressure variation with non-
porous solids, such as steel balls, as any leakage would cause pressure
variations and, accordingly, errors in permeability measurements of tight
porous samples (Heller et al., 2014). Before the data from porous samples were
analyzed, the leakage pressure from the steel ball experiment was subtracted
from the sample data to correct the modest (<5% of the pressure levels used
for permeability analyses) leakage effect.

The need for a unitary gas environment (a single gas used in both reference

and sample cells) is needed to successfully measure permeability via the GPT
method. The relative movement of gas molecules in the mass transfer process
is driven by the gas density gradient in the system. During gas transport, the
pressure variance was recorded and used to obtain the permeability coefficient.
However, when the gas in both cells is different, e.g., helium in the reference
and air in the sample cells, the mathematical analysis requires a complicated
correction accounting for the mean molar mass and the average gas dynamic
viscosity of the gas mixture. In this study, we present the calculation with the
viscosity of mixed gases for the GPT in the SI1. Since the mixed gas
environment is not recommended, air evacuation should be used for a well-
controlled unitary gas environment in the GPT.

A stable temperature is another critical point to ensure the success of the

GPT experiment. A sensitive pressure transducer in combination with the ideal
gas law, used to establish the relationship between pressure and gas volume
change, would be a much more convenient and precise way than the gas flow
meter to determine the gas permeability considering the measurement
accuracy. According to Amonton's law (Gao et al., 2004), the kinetic energy
of gas molecules is determined by the temperature, and any changes would
alter the molecular collision force causing a pressure variation and a
volumetric error. The GPT experiments were run two or three times on the
same sample, and the sample skeletal density at the end of the experiment were
obtained to check the overall indication of leakage and temperature control.
The experimental data with relatively large and stable skeletal density (mostly
the last run, from small but appreciable pressure change to reach stable values)
were used.

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

low permeability analyses of granular porous media: Mathematical solutions and
experimental methodologies.
