# Peer review of "A pulse-decay method for low permeability analyses of granular rock media"

_EGUsphere, 2022_

## Referee Comment (RC1)

**Review "A pulse-decay method for low permeability analyses of granular rock media"**
**By Zhang et al.**

**Scientific significance:**

The manuscript presents a comprehensive and meaningful evaluation of different solutions for determining the gas permeability of granular rock samples. The authors derive different solutions and discuss their application to different experimental parameters and their overall applicability. Their work provides a significant contribution for a detailed analysis of pressure (pulse) decay curves to derive the permeability of low permeable (nD) rocks.

**Scientific quality:**

In general, the quality is considered good. Their derivations of constitutive equations as well as analyses and parameter studies show important relations.

**Presentation quality:**

The manuscript is well-structured and mostly concise. Some redundancies and confusing sentence structures are listed below. Only one major concern is raised here: At multiple locations in the main manuscript, the authors are referring to the appendix. This is not only the case when referring to large mathematical derivations but also when referring to fundamental equations used for calculations of parameters analyzed. In addition, large parts of the experimental procedure and data are moved to the appendix; I consider this however relevant for understanding the application of the theory (the derivations of solutions to determine permeability). Many times, important information is denoted in parenthesis; for the reason of concision and readability, I suggest including the information in the sentence structure. To some extent, the text would benefit from proofreading with special attention to sentence structure and the use of English wording.

In conclusion, I consider moderate revisions by addressing the detailed comments below and by changing the structure of the manuscript in such a way that important parts of the appendix will be included to the main manuscript. In particular, important equations and figures should be shown to make the manuscript comprehensible to the reader.

**In-text comments:**

Line 23 – 25: "Nano-darcy level permeability measurements … are only practically feasible with gas invasion methods into granular-sized samples with short diffusion lengths and thereby reduced experimental duration;..." → This is a very strong and exclusive statement that I would not agree with. There are different (gas permeability) testing methods that work for low-perm materials. I suggest rephrasing the sentence in such a way: "Nano-darcy level permeability measurements … are frequently conducted with gas invasion methods into granular-sized samples with short diffusion lengths and thereby reduced experimental duration;..."

Line 70: "(e.g., 254 cm in diameter)" → I consider the additions info about sample size not relevant in this sentence since the size depends strongly on the experimental setup and might vary. If still needed I suggest using "consolidated cm-sized core-plug samples"

Line 81: "confounding" → is this the appropriate word here?

Line 92: It is somehow confusing where the mesh size/range comes from. Is there a conversion from mesh size to mm? Reader who are not familiar with the specs of this device might be confused.

Line 108: "The rest of this article is organized as follows." → redundand, can be deleted

Line 150: Reference error

Line 161 ff. Multiple times in the manuscript, the references are written twice.

Line 198: "… each solution holdS only …"

Line 208: " … helps TO select …"

Line 213 – 215: To my opinion, no need for recalling the definition of Kc again.

Line 221: delete "plotted in"

Line 223 & Fig. 1: the authors are referring to numbers not visible in the plot. I suggest to extend the x-axis to the point in the plot they are referring to.

Line 249: "…for Kc equals to 10,50 are… " → rephrase: "… where Kc Kc equals 10 or 50 are …"

Line 259: "… a minor difference but become very close …" → I don't entirely understand what is meant here. Please rephrase and carify.

Fig. 2(b): I suggest using a different color scheme since the light blue is hardly visible.

Line 291: Replace "happen" by "occur"

Line 297-298: "For example, … ". This is not a complete sentence; please, rephrase.

Line 302: "600s around for 0.1 nD" delete around

Line 305: I suggest removing the parentheses and adding the info to the sentence.

Line 327 ff: add space between equal sign and parameter/numbers

Line 334: "newLY"

Line 341-344: I suggest to move these sentences after the section header 4. (i.e. between lines 345 and 346)

Line 342: Be consistent in terminology. You used "mudstone" before

Line 352: "and or" → Confusing – please, rephrase.

Line 353: "See Fig. S2 in which it is shown how … " → Fig. S2 shows how…

Line 356: "… using different sample size from X-1. Similar pattern was observed for X-2 as well." → I would delete redundant wording and add the specific sample sizes; maybe like this: "… using different sample size ranging from XXX to XXX mm."

Line 359: To be consistent, you should use quotation marks around Penetration zone here as well.

Line 370: I strongly recommend to convert to SI units, or at least also denote SI units in parentheses.

Line 397 – 401: This paragraph appears a bit misplaced. I agree that measurements using liquids as permeating fluids underlie different assumptions and processes. But it appears here an explanation of why gases are used for testing. This section is, however, not the right place to state the usage of fluid type. I would simply omit the respective lines since the entire manuscript deals only with gas permeabilities.

Line 404: "behavior for a sample size of 0.675 mm (average granular diameter)" → change to "behavior for sample size with an average granular diameter of 0.675 mm"

Line 408: "For pressure range, argon … " → I don't understand what is meant here. Please, rephrase.

Line 424: "… has slow equilibrium time" → either change to "slow equilibrium process" or "long equilibrium time"

Line 429: "Adsorption… (Busch et al., 2008)" → This sentence appears incoherent. I would omit this sentence.

Line 439: "would provide more analyzable data to determine the …" → change to "would provide more data to be analyzed for determining the permeability"

Line 440 – 441: "This IS because… larger the pressure drop, AND (2) the longer …"

Line 442: "ThIS is consistent …"

Tab. 2: Change "size (mm)" to "granular size (mm)"

Line 466: "diameter smaller than Size A (average .. )" → Where does this term Size A come from (ssame for Fig. 8)? Please, change to "diameter smaller than on average 1.27 mm." please also apply changes to line 470.

Line 484: You tested the applicability of the derived solutions with experiments on mudstones but not for crystalline rocks (so far). Therefore, I would not mention crystalline rocks here.

Line 495: change to "criteria"

Line 527: I suggest rephrasing "…, one solution was valid for the early time when gas storage … and two were late-time solutions …"

Line 540: You are stating good repeatability. Did you test that? Did you show any proof of repeatability in your manuscript? If not, please delete the last part since it appears as speculation only.

---

## Community Comment (CC3)

Dear referee,

We express our sincere gratitude for the comments and valuable suggestions you've provided. In response to your feedback, we've made the following changes in the revised manuscript, especially for the paragraph starting in Line 185.

Comment #1. For the comments of "There are some vague statements about Cui et al. lacking "detailed analyses" of Kc and tau (in Line 187) and lacking a discussion of "practical applications" of their work (in Line 191), but a concise problem statement is lacking. So the novelty of the work is not clear to readers. I recommend revising this paragraph with a series of direct statements about specifically what was missing from Cui's work and how the present work addresses those shortcomings."

The whole paragraph starting in Line 187 has been revised as:
Based on diffusion phenomenology, Cui et al. (2009) presented two mathematical solutions similar to our Eqs. (3A) and (3C). In the work of Cui et al. (2009), however, one of late-time solution is missing, and error analyses are not provided. Besides, the lack of detailed analyses of $\tau$ and $K_c$ in the constitutive equations will likely deter the practical application of Eq. (3B), which is able to cover an experimental condition of small sample mass with a greater $\tau$ (further analyzed in Section 3). Furthermore, the early-time and late-time solution criteria are not analyzed, and the pioneering work of Cui et al. (2009) does not comprehensively assess practical applications of their two solutions in real cases, which is addressed in this study.

With respect to the clarification of similarities, differences, and innovations of our work, we acknowledge your concerns about the potential overlap with the study conducted by Cui et al. (2009). Both studies focus on the diffusion decay method for permeability measurements, investigating the same phenomena and employing the same method proposed by Cui et al. (2009). The work of Cui et al. (2009) presented two notable contributions: (1) they provided two solutions, one for early-time measurements and another for late-time measurements; and (2) they established a reference value of the storage capacity (Kc) as 50 and recommended the use of the late-time solution when Kc is greater than 50.

We acknowledge the excellent work performed by Cui et al. (2009), however, we believe their studies to be incomplete. Firstly, full mathematical solutions were not provided, as actually there are three solutions rather than two. Secondly, the mathematical error associated with each solution was not discussed, as all the three solutions are an approximation rather than an exact one. Most importantly, they overlooked the early-time solution, which is truly necessary, practical, and efficient for testing tight rock media.

In our work:
(1)  We provide a comprehensive and mathematically deduction process to obtain all three solutions, rather than directly borrowing the existing solutions from the realms of heat conduction and chemical diffusion.
(2)  We explain the classification of the early-time and late-time solutions, based on the

dimensionless time $\tau$ (never discussed by Cui et al.), and determined the specific value of 0.024 as the criterion from the exact solution calculated using MATLAB.

(3) Based on the discussion and classification of different $\tau$ values, we explain how the storage capacity (Kc) influences the solution selection.

(4) We conduct a kinetic analysis of several gas molecules and provide detailed analyses regarding sample mass, diameter, and equipment settings.

(5) We demonstrate the clear work-flow procedure for the application and selection of these three solutions that we've derived.

(6) Therefore, in the revised paragraph starting in Line 187, we emphasize that Cui et al. (2009) did not provide a comprehensive analysis of $\tau$ and Kc, and they overlooked the analyses of the early-time solution (Eq. 3C). In contrast, the solution being provided in current work (Eq. 3B) is practical for most common tight geomedia.

2. Regarding the comments: In Table 2, it would be helpful to have a comparison between results of the present work and those using Cui's methods to demonstrate improvement.

The comparison of these two solutions (from Cui et al.) and the third one has been added here (Second to last column and third to last column):

| Granular size (mm) | SMP-200 (nD) § | GPT test 1 (nD)£ | GPT test 2 (nD)£ | Average value (nD)£ | Fitting duration (s) | Solution type | Dimensionless time | Comparison for second solution (nD) | Comparison for third solution (nD) | Particle density (g/cm³) |
|---|---|---|---|---|---|---|---|---|---|---|
| 5.18 | - | 1.17 | 1.17 | 1.17 | 50-100 | ILT | 0.023-0.027 | **239 IET** | **1.31 LLT** | 2.631 |
| 2.03 | 14.2 | 0.45 | 0.41 | 0.43 | 50-100 | LLT | 0.026-0.028 | **11.1 IET** | **0.36 ILT** | 2.626 |
| 1.27 | - | 0.10 | 0.10 | 0.10 | 30-60 | LLT | CR* | **20.5 IET** | **0.09 ILT** | 2.673 |
| 0.67 | 0.65 | 0.08 | 0.04 | 0.06 | 30-60 | LLT | CR* | **1570 IET** | **0.03 ILT** | 2.658 |
| 0.34 | - | 0.002 | - | 0.002 | 30-60 | IET | CR* | **0.00076 LLT** | **0.00068 BLT** | 2.643 |

§ The results are from the SMP-200 using the GRI default method.

£ The results are from the GPT method we proposed.

* CR means the conflict results that the verified dimensionless time does not confirm the early- or late-time solutions using the solved permeability. For example, the verified dimensionless time would be > 0.024 using the early-time solution solved result and vice versa.

According to the method from Cui et al. (2009), they prefer using the late-time solution (ILT) for all the situations, while we provide the results accordingly with ILT, LLT, and IET and demonstrated here. The conclusion from the comparison is that: there is not much value difference between LLT and ILT method, which is the same as we analyzed in the error difference. However (1) the IET solution is necessary, (2) and LLT is more eurytopic, as discussed in this paper.

Overall, except the mathematical derivation, we deem the innovation and the significant improvement from this manuscript is the methodology and criteria for the practical utilization.

Thank you very much for your valuable feedback and suggestions.

---

## Author Response (AR1)

**Dear (Handling) editor,**

Based on feedback from two referees, we meticulously revised the manuscript, addressing each point raised one by one. Included here is a comprehensive description of our responses to their suggestions and concerns. In particular, regarding the concerns of the second referee, we firmly believe in the innovation of our work and have highlighted its novelty specifically here. We also undertook an additional thorough review of the manuscript including the formulas, we revised a few typos.

Furthermore, our gratitude goes to AE Monica Riva for her diligent handling and effort in inviting over 20 referees.

Thank you.

**Dear referee #1,**

We express our gratitude for the thorough review and insightful remarks on our work, particularly a total of 42 detailed comments addressing punctuation, word choice, sentence structure, and overall organization of this manuscript. We have carefully reviewed each comment and made the corresponding revisions accordingly, which improve the presentation and clarity of the work. Thank you!

- 1. Line 23 25: Nano-darcy level permeability measurements ... are frequently conducted with gas invasion methods into granular-sized samples with short diffusion lengths and thereby reduced experimental duration;..."
- 2. Line 70: we revised the "e.g., 254 cm in diameter" to be "consolidated cm-sized coreplug samples"
- 3. Line 81: we revised the "confounding" effects as the "side" effects.
- 4. Line 92: 10-60 mesh has been explained specificity to be 0.67 mm to 2.03 mm.
- 5. Line 108: "The rest of this article is organized as follows." has be deleted in the revised manuscript.
- 6. Line 150: Reference error has been corrected.
- 7. Line 161 The second reference has been deleted.
- 8. Line 198: hold has been corrected as holds.
- 9. Line 208: " ... helps TO select ... "
- 10. Line 213 215: The second explanation of Kc has been deleted.
- 11. Line 221: delete "plotted in"
- 12. Line 223 & Fig. 1: We have extended the x-axis to the point in the plot of 50.

- 13. Line 249: We have rephrased the sentence as "... where Kc equals 10 or 50 are..."
- 14. Line 259: "... a minor difference but become very close ..." This sentence has been revised as " there difference is very small especially for  $K_c > 10$ ".
- 15. Fig. 2(b): We've used different color schemes to enable it more visible.
- 16. Line 291: "Happen" has been revised as "occur".
- 17. Line 297-298: "For example" has been revised as "This is particularly noticeable for".
- 18. Line 302: for "600s around for 0.1 nD", the word of "around" has been deleted.
- 19. Line 305: The info in the parentheses has been rephrased as "The mudrock samples we tested, with results presented in Section 5.3, exhibit low permeabilities, approximately on the order of 0.1 nD."
- 20. Line 327 ff: add space between equal sign and parameter/numbers. This has been corrected for the whole paragraph (from Lines 326-332).
- 21. Line 334: The word "new" has been revised as "newly".
- 22. Line 341-344: We have relocated the paragraph from Lines 341-344 to the suggested position between Lines 345 and 346; thanks for the comment.
- 23. Line 342: We've unified the term as mudstone.
- 24. Line 352: We've rephrased as "or".
- 25. Line 353: We've rephased as "Fig. S2 shows how".
- 26. Line 356: The sentence has been simplified as "Fig. 5 shows the pressure variance with time during the experiment using sample size from 0.34 mm to 5.18 mm for sample X-1 and sample X-2."
- 27. Line 359: Quotation marks have been applied to "Penetration zone" here.
- 28. Line 370: Psi has been converted to SI units in parentheses, and 50 and 200 psi is 0.345 MPa to 1.38 MPa.
- 29. Line 397 401: We've deleted the content between Lines 397-401 of "Though the liquid permeability is not complicated by the gas slippage effect, the liquid test is difficult in achieving the flow state of Knudsen number greater than  $10^{-3}$ , which normally occurs in the ultra-low permeability media. Therefore, gases are chosen, and practically needed, as the testing fluid in this work."
- 30. Line 404: We have rephrased the "behavior for a sample size of 0.675 mm (average granular diameter)" to "behavior for sample size with an average granular diameter of 0.675 mm".
- 31. Line 408: About "For pressure range, argon ... ", this sentence has be revised as "In terms of pressure drop, argon exhibited the most significant decrease."
- 32. Line 424: For "slow equilibrium time", we've changed as "slow equilibrium process".
- 33. Line 429: "Adsorption... (Busch et al., 2008)"; this sentence has been deleted.
- 34. Line 439: "would provide more analyzable data to determine the …" has been revised as "would provide more data to be analyzed for determining the permeability".

- 35. Line 440 441: We've added "is" and "and" in this sentence.
- 36. Line 442: "The" has been revised as "This".
- 37. Tab. 2: "size (mm)" has been changed as "granular size (mm)"
- 38. Line 466: The description related to the Size A has been replaced by the value of 1.27 mm.
- 39. Line 484: We've deleted the crystalline rocks here.
- 40. Line 495: We've added the description that 2 mm is the criteria.
- 41. Line 527: We've revised sentence to improve clarity "Of the three derived solutions, one is valid during early times when the gas storage capacity  $K_c$  approaches infinity, while the other two are late-time solutions valid when  $K_c$  is either small or tends towards infinity."
- 42. Line 540: We tested each of the samples twice, and the results were demonstrated in Table 2, from the results there, we think the data demonstrated a good repeatability.

**Dear referee #2,**

We express our sincere gratitude for the comments and valuable suggestions you've provided. In response to your feedback, we've made the following changes in the revised manuscript, especially for the paragraph starting in Line 185.

Comment #1: For the comments of "There are some vague statements about Cui et al. lacking "detailed analyses" of  $K_c$  and tau (in Line 187) and lacking a discussion of "practical applications" of their work (in Line 191), but a concise problem statement is lacking. So the novelty of the work is not clear to readers. I recommend revising this paragraph with a series of direct statements about specifically what was missing from Cui's work and how the present work addresses those shortcomings."

The whole paragraph starting in Line 187 has been revised as:

Based on diffusion phenomenology, Cui et al. (2009) presented two mathematical solutions similar to our Eqs. (3A) and (3C). In the work of Cui et al. (2009), however, one of late-time solution is missing, and error analyses are not provided. Besides, the lack of detailed analyses of  $\tau$  and  $K_c$  in the constitutive equations will likely deter the practical application of Eq. (3B), which is able to cover an experimental condition of small sample mass with a greater  $\tau$  (further analyzed in Section 3). Furthermore, the early-time and late-time solution criteria are not analyzed, and the pioneering work of Cui et al. (2009) does not comprehensively assess practical applications of their two solutions in real cases, which is addressed in this study.

With respect to the clarification of similarities, differences, and innovations of our work, we acknowledge your concerns about the potential overlap with the study conducted by Cui et al. (2009). Both studies focus on the diffusion decay method for permeability measurements, investigating the same phenomena and employing the same method proposed by Cui et al.

(2009). The work of Cui et al. (2009) presented two notable contributions: (1) they provided two solutions, one for early-time measurements and another for late-time measurements; and (2) they established a reference value of the storage capacity ( $K_c$ ) as 50 and recommended the use of the late-time solution when  $K_c$  is greater than 50.

We acknowledge the excellent work performed by Cui et al. (2009), however, we believe their studies to be incomplete. Firstly, full mathematical solutions were not provided, as actually there are three solutions rather than two. Secondly, the mathematical error associated with each solution was not discussed, as all the three solutions are an approximation rather than an exact one. Most importantly, they overlooked the early-time solution, which is truly necessary, practical, and efficient for testing tight rock media.

In the revised manuscript during the lines of 203 to 212, we clarified that: the influence of parameters  $K_c$  and  $\tau$  on the solution of constitutive equation is analyzed and a specific value of dimensionless time ( $\tau = 0.024$ ) is proposed as the criterion required to detect the early-time regime from the late-time one for the first time in the literature. We also demonstrate that the early-time solution of Eq. (3C), which has been less considered for practical applications in previous studies, is also suitable and unique under common situations. Besides, the error of the approximate solution compared to the exact solution and their capabilities are discussed, as it helps to select an appropriate mathematical solution at small  $\tau$  values. Moreover, we showcase the unique applicability and feasibility of the new solution of Eq. (3B).

We conclude the novelty of our research in the following detailed aspects:

- (1) We provide a comprehensive and mathematical deduction process to obtain all three solutions, rather than directly borrowing the existing solutions from the realms of heat conduction and chemical diffusion. The entire derivation enables us to disclose the impact of various parameters (mainly the  $K_c$  and  $\tau$ ) on the configuration of the testing system. Additionally, it illustrates the error analysis and the data fitting procedure. The robust mathematical derivation laid the foundation to improve this method to a higher level.
- (2) We explain the classification of the early-time and late-time solutions, based on the dimensionless time  $\tau$ , and concluded the specific value of  $\tau$ =0.024 as the criterion using the exact solution curve from MATLAB. This criterion has never been proposed and discussed by previous studies.
- (3) Based on the discussion and classification of different  $\tau$  values, we then explain how the storage capacity ( $K_c$ ) influences the solution selection in a specific way. We proposed the dimensionless density of  $K_f$  in section 3.1, and discussed the
- (4) We conduct a kinetic analysis of several gas molecules and provide detailed analyses regarding sample mass, diameter, and equipment settings. We demonstrate the clear workflow procedure for the application and selection of these three solutions that we've derived. We update our work to be a more mature and accessible standard procedure, which is more easily for utilization.
- (5) We discussed and proved the practical utilization of the new derived equation.
- (6) Therefore, in the revised paragraph starting in Line 187, we emphasize that Cui et al. (2009) did not provide a comprehensive analysis of  $\tau$  and  $K_c$ , and they overlooked the analyses

of the early-time solution (Eq. 3C). In contrast, the solution being provided in current work (Eq. 3B) is practical for most common tight geomedia.

Comment #2: In Table 2, it would be helpful to have a comparison between results of the present work and those using Cui's methods to demonstrate improvement.

Granular Unselected SMP-200 Fitting GPT test GPT test Dimensionl Particle density Average size Solution (nD) § 1 (nD)£  $2 (nD)^{f}$ value (nD)£ duration (s) ess time  $(g/cm^3)$ (mm)  $(nD)^*$ 239(IET) 5.18 -1.17 1.17 1.17(ILT) 50-100 0.023-0.027 2.631 1.31(LLT) 11.1(IET) 2.03 14.2 0.45 0.41 0.43(LLT) 50-100 0.026-0.028 2.626 0.36(ILT) 20.5(IET) 1.27 0.10 0.10 0.10(ILT) 30-60  $CR^*$ 2.673 \_ 0.09(ILT) 1570(IET) 0.65 0.67 0.08 0.04 0.06(LLT)  $CR^*$ 30-60 2.658 0.03(ILT) 0.00076(LLT) 0.34 0.02(IET) 2.643 0.02 30-60  $CR^*$ 0.00068(BLT

The comparison of these two solutions (from Cui et al.) and the third one has been added here (Unselected Solution):

§ The results are from the SMP-200 using the GRI default method.

£ The results are from the GPT method we proposed.

 $^{*}$  CR means the conflict results that the verified dimensionless time does not confirm the early- or late-time solutions using the solved permeability. For example, the verified dimensionless time would be > 0.024 using the early-time solution solved result and vice versa.

\* represents the result which failed for the criteria of dimensionless time

According to the method from Cui et al. (2009), they prefer using the late-time solution (ILT) for all the situations, while we provide the results accordingly with ILT, LLT, and IET and demonstrated here. The conclusion from the comparison is that: there is not much value difference between LLT and ILT method, which is the same as we analyzed in the error difference. However (1) the IET solution is necessary, (2) and LLT is more eurytopic, as discussed in this paper of section 3.2 and 3.4.

Overall, except the mathematical derivation, we deem the innovation and the significant improvement from this manuscript is the methodology and criteria for the practical utilization.

Thank you very much for your valuable feedback and suggestions.